# Allelic expression mapping across cellular lineages to establish impact of non-coding SNPs

Veronique Adoue[1], Alicia Schiavi[2,†], Nicholas Light[2,†], Jonas Carlsson Almlöf[3], Per Lundmark[3], Bing Ge[2], Tony Kwan[2], Maxime Caron[2], Lars Rönnblom[4], Chuan Wang[3], Shu-Huang Chen[2], Alison H Goodall[5,6,7], Francois Cambien[7,8], Panos Deloukas[7,9], Willem H Ouwehand[7,10,11], Ann-Christine Syvänen[3] & Tomi Pastinen[2,*]

## Abstract

Most complex disease-associated genetic variants are located in non-coding regions and are therefore thought to be regulatory in nature. Association mapping of differential allelic expression (AE) is a powerful method to identify SNPs with direct *cis*-regulatory impact (*cis*-rSNPs). We used AE mapping to identify *cis*-rSNPs regulating gene expression in 55 and 63 HapMap lymphoblastoid cell lines from a Caucasian and an African population, respectively, 70 fibroblast cell lines, and 188 purified monocyte samples and found 40–60% of these *cis*-rSNPs to be shared across cell types. We uncover a new class of *cis*-rSNPs, which disrupt footprint-derived *de novo* motifs that are predominantly bound by repressive factors and are implicated in disease susceptibility through overlaps with GWAS SNPs. Finally, we provide the proof-of-principle for a new approach for genome-wide functional validation of transcription factor–SNP interactions. By perturbing NFκB action in lymphoblasts, we identified 489 *cis*-regulated transcripts with altered AE after NFκB perturbation. Altogether, we perform a comprehensive analysis of *cis*-variation in four cell populations and provide new tools for the identification of functional variants associated to complex diseases.

**Keywords** allelic expression; *cis*-rSNPs; complex disease; NFκB; repressor
**Subject Categories** Genome-Scale & Integrative Biology; Chromatin, Epigenetics, Genomics & Functional Genomics
**Mol Syst Biol. (2014) 10: 754**

## Introduction

The vast majority of Genome-Wide Association Studies (GWAS) variants for complex diseases lie in non-coding DNA (~90%) and are specifically enriched in areas of open chromatin in cell types that are relevant to the disease of interest (Manolio *et al*, 2009; Maurano *et al*, 2012). These non-coding variants are thought to act primarily through altering regulation of gene expression in *cis*. Characterization and prediction of the cell-type specificity of *cis*-regulatory variation are therefore important in identifying causal disease-relevant *cis*-rSNPs (Pastinen, 2010). To date, most studies investigating *cis*-regulatory variation have utilized expression quantitative trait loci (eQTL) mapping, where variants are tested for their association with gene expression (Schadt *et al*, 2003). While eQTL studies have proven to be a powerful tool in investigating the genetics of gene expression in a broad sense, the investigation of *cis*-regulatory mechanisms requires an approach that isolates the role of *cis*-rSNPs to transcription. The *cis*-acting components of expression variation can be identified through the mapping of differences in allelic expression (AE), which is the measure of relative expression between two allelic transcripts (Ge *et al*, 2009). The parallel genotyping of genomic DNA and RNA (cDNA) on high-density genotyping chips allows interrogation of AE variation across transcribed loci, including both exons and introns. Since both alleles are impacted by the same trans-acting and environmental effects, AE mapping reduces the complexity of gene expression to its *cis* components. A recent study showed that this approach greatly improved the sensitivity of detecting *cis*-rSNPs compared to standard eQTL mapping and demonstrated that an eightfold decrease in sample size

1  Institute National de la Santé et de la Recherche Médicale (INSERM), U1043, Toulouse, France
2  Department of Human Genetics, McGill University and Genome Quebec Innovation Centre, Montreal, QC, Canada
3  Department of Medical Sciences, Molecular Medicine, Science for Life Laboratory, Uppsala University, Uppsala, Sweden
4  Rheumatology, Department of Medical Sciences, Uppsala University, Uppsala, Sweden
5  Department of Cardiovascular Science, University of Leicester, Leicester, UK
6  Leicester NIHR Biomedical Research Unit in Cardiovascular Disease, Glenfield Hospital, Leicester, UK
7  Cardiogenics Consortium
8  INSERM UMRS 937, Pierre and Marie Curie University and Medical School, Paris, France
9  Wellcome Trust Sanger Institute, Wellcome Trust Genome Campus, Cambridge, UK
10 Department of Haematology, University of Cambridge, Cambridge, UK
11 National Health Service Blood and Transplant, Cambridge Centre, Cambridge, UK
   *Corresponding author. Tel: +1 514 398 1777; E-mail: tomi.pastinen@mcgill.ca
   †These authors contributed equally to this work

is sufficient to achieve the same statistical power as in standard eQTL mapping (Almlof et al, 2012). Studies of AE mapping in lymphoblastoid cell lines (LCLs) have revealed that approximately 30% of all loci have significant AE imbalance, with cis-rSNPs explaining more than 50% of the population variance in AE (Ge et al, 2009). The effect of cis-rSNPs on other disease-relevant cell types depends on the proportion of regulatory elements that are shared between cell types, compared to those that are specific to a single cell type or restricted to a small subset of cell types. Earlier eQTL studies have suggested that over 50% of cis-rSNPs are shared between any two tissues, for example, LCLs and fibroblasts (Emilsson et al, 2008; Kraft, 2008; Dimas et al, 2009; Ding et al, 2010).

Large-scale functional mapping projects, such as ENCODE, have generated massive collections of high-resolution functional genomics data. However, much of this information has yet to be integrated with studies on population expression variation (The ENCODE Project Consortium, 2012). Hundreds of new transcription factor (TF) recognition motifs that exhibit cell-selective occupancy were discovered using DNase I footprinting (Neph et al, 2012), providing the opportunity to study new DNA elements in conjunction with population variation. Importantly, these projects have reported a large fraction of open and functional chromatin sites to be cell-type specific. This is in contrast with previous eQTLs reports, which showed considerable sharing in functional regulatory variation across tissues (Emilsson et al, 2008; Kraft, 2008; Dimas et al, 2009; Ding et al, 2010; The ENCODE Project Consortium, 2012; Thurman et al, 2012).

Despite progress in mapping functional elements, defining causal cis-rSNPs among correlated sites in high linkage disequilibrium (LD) remains a challenge. Traditional tools such as reporter gene assays typically isolate the putative regions from their functional chromatin context (Cirulli & Goldstein, 2007). New approaches for the global functional assessment of the molecular bases of mapped associations are needed.

This study utilizes the allelic expression approach to investigate the genetic determinants of differential allelic expression of protein-coding and non-coding genes across four cell populations. We examine TF binding site disruption by mapped cis-rSNPs and investigate their regulatory role on gene expression across tissues. Finally, we propose a novel platform to globally examine the role of key regulators by combining allelic expression read-outs with targeted approaches to perturb TFs in living cells.

# Results

### Quantitative allelic expression measurements and mapping in 4 cell populations

Genome-wide quantitative AE measurements were carried out on Human1M-Duo BeadChips (Ge et al, 2009) for four populations covering 3 distinct cell types. As in our previous work, we used both intronic and exonic SNPs passing the signal intensity threshold, with 75% of the SNPs used for AE mapping located intronically in non-processed transcripts. We restricted our analysis to differences in normalized allele ratios in RNA (cDNA) at heterozygous sites averaged across fully annotated primary transcripts (Grundberg et al, 2011), in order to detect allelic differences impacting full transcripts,

rather than changes in splicing or 3′ usage (Ge et al, 2009) (see Methods). In addition to 55 HapMap lymphoblastoid cell lines (LCLs) from a Caucasian population (CEU), we included 63 LCLs from an African population (YRI), 70 fibroblast cell lines from a Caucasian population (FBs) (Wagner et al, 2014), and 188 purified monocyte samples (MNCs) from unrelated healthy donors residing in the UK (Almlof et al, 2012). This selection of cell types enabled us to capture a wide range of potential cis-variants and aided in the fine mapping of common variants between populations.

The application of the BeadChip genotyping process, which includes amplification, allows for the detection of rare transcripts. In order to focus on genes with biologically relevant expression levels, we restricted our analysis to expressed loci independently determined using RNA-seq expression data, with up to eight samples per cell type (see Methods). Using this method, we identified 11,723, 9,982, and 11,487 non-overlapping expressed transcripts in LCLs, fibroblasts, and monocytes, respectively. We next applied a filter for the statistical significance of genetic effects on allelic expression in order to limit the discovery of false-positive associations, requiring that loci be mapped below the threshold of 1% FDR (see Methods). This led to the detection of 49, 36, and 81% of allelically regulated transcripts in LCLs, fibroblasts, and monocytes, respectively. Examples of proximal and distal allelic expression associations in individual transcripts are depicted in Fig 1. In order to include the optimal number of associated SNPs in our analysis, we assessed simulated candidate loci with known "causal" sites (Supplementary Methods). Through the simulation analysis, we observed that the percentage of missing causal SNPs is below 5% when the top 10 ranked SNPs by P-value are included per locus, and thus, we focused our subsequent analyses using this cutoff (Supplementary Fig S1). A summary of mapped associations and number of tested SNPs are displayed in Table 1 and Supplementary Table S1, respectively.

All classes of transcripts annotated in GENCODE V15 were included in our analysis. This allowed us to map cis-regulatory variants for 266 lincRNAs, 642 processed transcripts, 308 antisense transcripts, and 15 sense-intronic transcripts (Supplementary Table S2).

### Shared cis-regulatory variation is high between cell types and populations

In order to assess the biological relevance of cis-regulatory variation between the four cell population panels, we defined cell-type-specific and shared associations. We applied a stringent approach to assess exactly shared top associations across tissues requiring not only significant association in both tissues, but also converging association pattern. To account for cases of detecting true cis-regulatory associations at weak significance levels, which could result in underestimating the number of shared associations, we used a method that is conceptually similar to that used by Nica et al (2011). For each locus, all primary associations in one population were compared to the first percentile of mapped SNPs in another cell population (see Methods). We observed 10–23% of SNP-transcript associations in each tissue (same population) as shared in the two others and that a majority is shared between at least two tissues (Table 1 and Supplementary Table S3). Monocytes showed the highest (61%) proportion of cell-selective associations. We also identified 38–48% of loci shared between different ethnicity (CEU

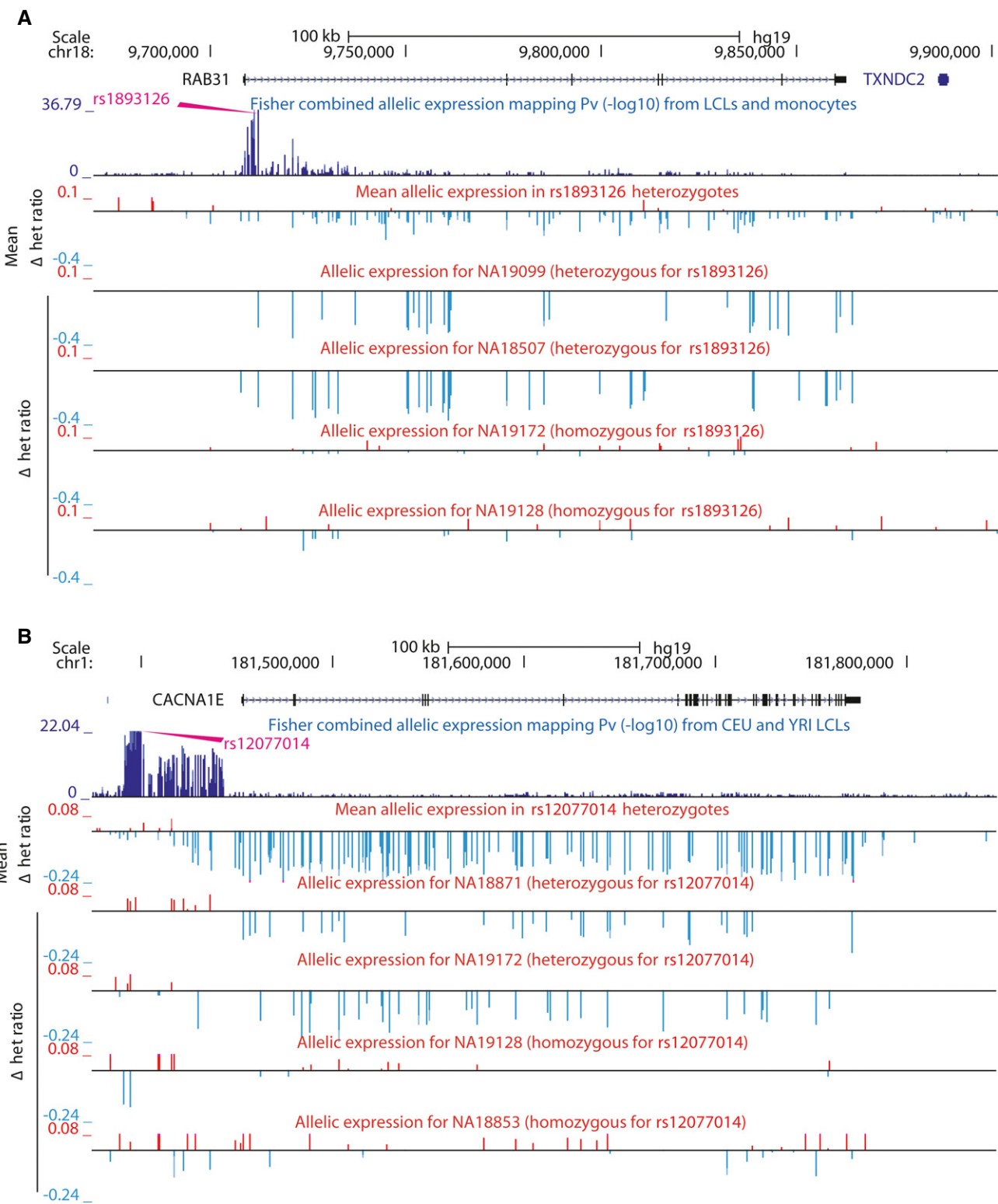

**Figure 1. Examples of allelic expression mapping in individual transcripts.**

Regression tests were carried out using phased AE data: *y*-axis shows the mean Δ het ratio across all individuals heterozygous for top SNP across population or the Δ het ratio for individual examples. NA19099 and NA18507 in panel (A) and NA 18871 and NA19172 in panel (B) are heterozygous for top SNP, and profiles reflect differential AE at population level. NA19172 and NA19128 in panel (A) and NA19128 and NA18853 in panel (B) are homozygous for top SNP and show weak differential AE with no significant bias toward one haplotype.

A  Example of *cis*-rSNP located close to the TSS of its associated transcript. Differential AE of *RAB31* is associated to *rs1893126* genotype.

B  Example of *cis*-rSNP located far away from its associated transcript. The strongest association for *CACNA1E* is *rs12077014* and is located > 53 kb from its TSS.

**Table 1.  Number of mapped loci per gene type and cell population.**

| | Cell Types | | | |
|---|---|---|---|---|
| | Lymphoblastoid Cell Lines-CEU | Lymphoblastoid Cell Lines-YRI | Fibroblasts | Monocytes |
| Number of samples | 55 | 63 | 70 | 188 |
| Total number of associations | 3343 | 3982 | 3983 | 8175 |
| Number of associations per gene type | | | | |
| Protein-coding | 2938 | 3486 | 3526 | 7367 |
| lincRNA | 74 | 109 | 89 | 156 |
| Processed transcripts | 200 | 241 | 234 | 437 |
| Antisense transcripts | 108 | 118 | 115 | 162 |
| Sense-intronic transcripts | 6 | 6 | 3 | 11 |
| Others | 17 | 22 | 16 | 42 |
| Cell-selective associations[a] | 1293 (39%) | 1997 (50%) | 1610 (40%) | 4978 (61%) |
| Shared across all tissues[b] | 613 (23%) | – | 668 (19%) | 769 (9.5%) |
| Shared across all tissues and populations[c] | 407 (15%) | 371 (12%) | 435 (13%) | 485 (6%) |

[a]SNP–transcript associations mapped in one cell population.
[b]SNP–transcript associations mapped in CEU LCLs, fibroblasts, and monocytes (same population).
[c]SNP–transcript associations mapped in CEU LCLs, YRI LCLs, fibroblasts, and monocytes.

versus YRI LCLs) of the same cell type (Supplementary Table S3). We noted that this tissue sharing is conservative as compared to methods recently used to estimate tissue sharing in eQTL studies allowing for uncertainty in mapping accuracy (Grundberg *et al*, 2012). Estimates of pairwise tissue sharing based on $\pi1$ values are similar to eQTL studies and range from 39% between YRI LCLs and fibroblasts to over 60% for monocyte lead associations and the three other sample panels (Supplementary Table S4) (see Methods). We explored the relevance of mapped shared associations in full datasets using IPA (Ingenuity Systems, www.ingenuity.com). Networks of genes associated to *cis*-rSNPs shared across all three studied cellular lineages are involved in basic cellular functions (Supplementary Fig S2). In contrast, associations shared by the two LCL panels are enriched in cell-to-cell signaling and immune response networks, and associations shared between lymphoblasts and monocytes are enriched in networks related to immunological disease and immune response. Finally, we hypothesized that the variation in the number of mapped loci, which are tissue independent or cell-type selective, could be due to differences in the number of analyzed samples, leading to discovery of weaker effects less likely to be shared across tissues in larger samples. Down sampling to equally powered datasets (see Supplementary Methods) shows the ratio of cell type-dependent versus all associations is quite stable across comparisons, suggesting that the level of sharing we observed is mainly influenced by (i) tissue-dependent differences, where cultured cells of Caucasian origin (CEU and FB) show lowest level of tissue differentiation and (ii) population genetic variation (YRI versus all others) where sequence divergence predominates (Supplementary Fig S3).

To improve the mapping resolution, we next applied meta-analyses across populations (see Methods). By breaking up blocks of SNPs in high LD, this approach significantly reduced the total number of associated SNPs per loci by ~2.1-fold (chi-squared test, $P < 2 \times 10^{-32}$, Supplementary Fig S4) (Hess & Iyer, 2007). Intersecting our data with the RegulomeDB (RegDB) database (Boyle *et al*,

2012) yielded a significant enrichment of mapped SNPs overlapping functional elements (1.1-fold, chi-squared test, $P = 0.02$). A significant increase in the proportion of GWAS hits (1.9-fold, chi-squared test, $P = 2 \times 10^{-16}$) was also observed, particularly in autoimmune diseases (1.5-fold, chi-squared test, $P = 1.4 \times 10^{-32}$). These findings may be partially attributed to the use of two immune cell types (lymphoblasts and monocytes) among the three cell lineages studied. The spatial distribution of the mapped *cis*-rSNPs follows an expected trend (Nica *et al*, 2011): high density of sites in 5′UTRs, gene bodies, and 3′UTRs, which rapidly decreases as a function of the distance from the gene in flanking regions. The associations shared between cell populations display a striking enrichment at the TSS of the associated transcript. In parallel, we observe a depletion of cell-type-specific associated variants at the TSS, with a simultaneous increase in more distal associations (Supplementary Fig S5). Only 0.4% of all rSNPs associations localized further than 200 kb away from the associated transcript. The density of long-range associations may be slightly underestimated for the monocyte sample due to reliance on statistical rather than family-based approach in phase assignment and potential of confounding errors in long-range haplotypes. We observe rate of long-range effects of 0.2% in monocytes versus 1.1% in the three other cell types. Overall, these data support the validity of our approach to map high-quality *cis*-regulatory variants in different cell population.

We further observed the same *cis*-rSNP to be associated to AE of multiple transcripts (up to 10) in the same cell population for 35% of all mapped transcripts ($n = 3825$). Moreover, we detect the opposite allelic direction of regulatory effect for transcripts linked to the same *cis*-rSNPs in 33% of these cases. Two examples of these complex loci are depicted in Supplementary Fig S6. We previously reported this type of effect for a single disease locus (Verlaan *et al*, 2009) and showed an impact of genetic variants on higher order chromatin function. Supplementary investigation would be needed to establish if this could be a common phenomenon across human cell types.

## Cis-rSNPs are linked with disease variants

To investigate the role of mapped regulatory variants and disease, *cis*-rSNPs for each cell-type specificity class were intersected with hits from the NHGRI Catalog of Published GWAS at genome-wide significance ($P < 5 \times 10^{-8}$). In total, we identified 540 loci with at least one *cis*-rSNP in nearly absolute LD ($r^2 \geq 0.9$) with a disease hit (Table 2; Supplementary Table S5). These *cis*-rSNPs showed a high degree of functionality, with 42.2% of them falling into categories 1–5 from RegDB, which is significantly higher than for all *cis*-rSNPs in our study (34.8%) (chi-squared test, $P = 3.8\text{E} \times 10^{-23}$). Transcripts for non-coding RNAs, including lincRNA, antisense, processed and sense-intronic RNAs, represent 7.8% ($n = 42$) of the disease-associated loci. One interesting example is located in the genomic region 3q12.3, where *rs771767*, a SNP linked by GWAS to multiple sclerosis, localizes to an intergenic region > 168 Kb from the closest protein-coding gene (Fig 2A–B) (International Multiple Sclerosis Genetics C, Wellcome Trust Case Control C, 2011). This variant is located in a region of open chromatin (ENCODE data) and, in our data, is the top-ranked *cis*-rSNP ($P = 1.96 \times 10^{-6}$) for the lincRNA *RP11-221J22.1*. This association is specific to monocytes, a cellular lineage with a role in instigating neuroinflammation in multiple sclerosis disease models (Hendriks *et al*, 2005). Globally, we found a correlation between the categories of traits of the GWAS hits and the cell-type specificity of the *cis*-rSNPs with which they are in LD (Fig 2C). SNPs associated to hematological traits are significantly enriched (1.9-fold, chi-squared test, $P < 0.01$) in *cis*-rSNPs mapped in monocytes. We also observed a significant enrichment (chi-squared test, $P < 0.05$) of variants associated with auto-immune diseases among the *cis*-rSNPs mapped in the immune-related cell types, the lymphoblasts (1.6-fold), and the monocytes (1.4-fold), as has been earlier observed in eQTL studies (Fairfax *et al*, 2012). Finally, we identified shared genetic effects between autoimmune diseases as previously described (Cotsapas *et al*, 2011). One key example is located in 9q34.3: *rs3812560* (RegDG score = 1f), which is linked to four different diseases, and is associated to differential allelic expression of at least five genes (Supplementary Fig S7), suggesting that *rs3812560* may act as a master regulator, involved in multiple physiological contexts and diseases. Molecular investigation of the role of *rs3812560* and other master regulators may be beneficial in the elucidation of the shared mechanisms involved in the development of autoimmune diseases.

**Table 2.  Number of loci with a *cis*-rSNP in high LD with SNP from the GWAS catalog.**

|  | Number of loci with *cis*-rSNP in high LD[a] with a GWAS hit (LD $\geq$ 0.9) | Number of locus–disease associations[b] |
| --- | --- | --- |
| Cell-selective[b] | 360 | 418 |
| Shared by at least 2 cell types | 182 | 230 |
| Shared by all cell types | 20 | 25 |
| Total | 540 | 648 |

[a]$r^2 > 0.9$
[b]As a same *cis*-regulatory variant can be associated to multiple diseases, the total number of locus–disease associations are also reported.

## Motif disruption at *cis*-rSNPs revealed new regulatory function

One mechanism by which *cis*-rSNPs may act and mediate disease risk is through the disruption of transcription factor binding sites. Databases of TF binding matrices, such as TRANSFAC (http://www.biobase-international.com), can be used to detect these events. However, they contain only a minority of the human transcription factors with predicted high-quality sequence-specific DNA binding domains (Vaquerizas *et al*, 2009). Recently, Neph *et al* (2012) performed unbiased *de novo* motif discovery within the footprints left by regulatory factor binding to genomic DNA and protecting the underlying sequence from cleavage by DNase I in 41 cell types. This approach allowed them to discover 683 unique *de novo* motifs (numbered 1–683), 289 of which showed no match in previous motif databases. This new genome-wide dataset of transcription factor binding, in conjunction with the major motif database TRANSFAC which contains 721 motifs, provides an opportunity to interrogate both known and unknown DNA–protein interactions that may be affected by *cis*-rSNPs.

To investigate the relationship between *cis*-rSNPs and TF binding, we used the FIMO motif scanning software (see Methods) to calculate the number of disrupted binding sites per motif at *cis*-rSNPs based on cell-type specificity (Supplementary Table S6). As expected, sites for general transcription factors such as SP1, AP1/2, and CTCF are frequently disrupted in all cell types (Supplementary Table S7). However, we also observed cell-type-specific activity at TRANSFAC and footprint-derived motifs (Fig 3) (chi-squared test, $P < 0.05$): For example, binding motifs for NF$\kappa$B and IRF factors and the *de novo* motif "616" are frequently disrupted in LCLs in agreement with previous studies (Ernst *et al*, 2011; Neph *et al*, 2012). Motifs for FOXO3A and PU.1 factors are more specific to fibroblasts and monocytes (Ito *et al*, 2009; Wang *et al*, 2014).

We next explored the regulatory role of factors binding TRANSFAC and footprint-derived motifs by combining allele-specific matrix affinity scores at *cis*-variants with the AE data of the associated transcript. Motifs for transcriptional activators are expected to exhibit a higher matrix score for the same haplotype as the overexpressed allele, while inhibitors should more frequently recognize the haplotype for the under-expressed allele. Using genotypes at *cis*-variants from the four population panels, we systematically compared the matrix score between both alleles and across all sites where a given motif was found. We identified 63 motifs (11%) with significant matrix score allelic bias ($P$ binomial test < 0.01) (Table 3 and Supplementary Table S8). Among these, 41 (65%) were associated with an activatory activity: We observed significantly more cases where the "higher matrix score" was on the same haplotype as the more expressed allele. The vast majority of these potential activator-binding sites overlapped a known motif from TRANSFAC (76%). The most significant of these motifs bind well-known transcriptional activators: *NF$\kappa$B*, *CEBP* (*CCAAT*/enhancer-binding family), and *PU1*, a lymphoid-specific enhancer (Kueh *et al*, 2013). Overall, nine "*de novo*" motifs present potential enhancer activity ($P$ binomial test < 0.01).

We identified 22 motifs with significantly more cases where the "higher matrix score" was on the same haplotype as the expected under-expressed allele, suggesting a repressor activity. Several of the top 10 most significant motifs are bound by factors with published inhibitory activity, such as *NRSF* (a.k.a. *REST*) (Chong

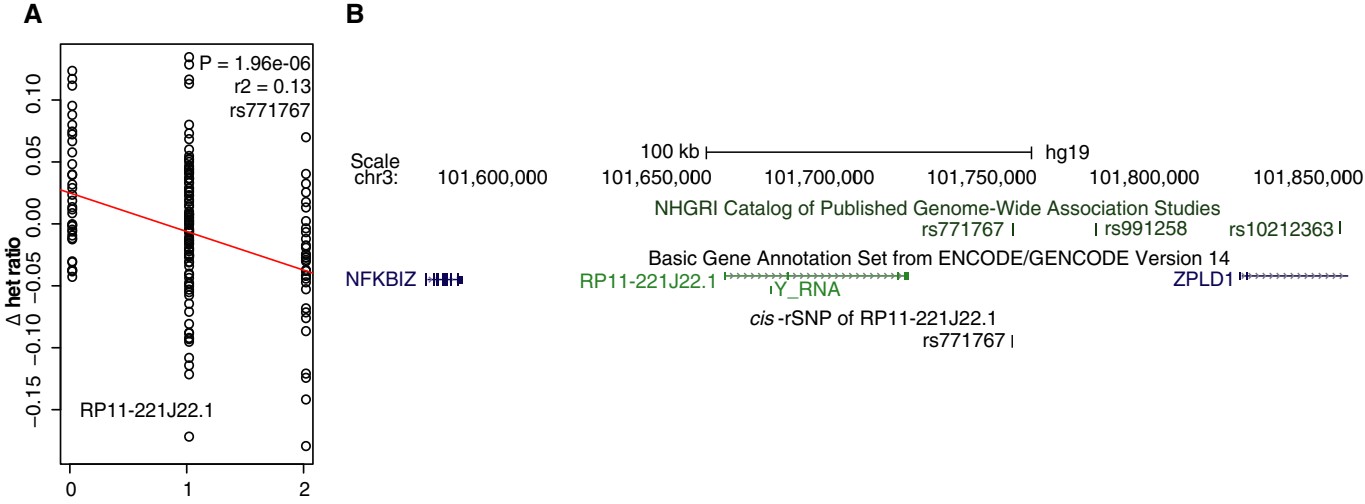

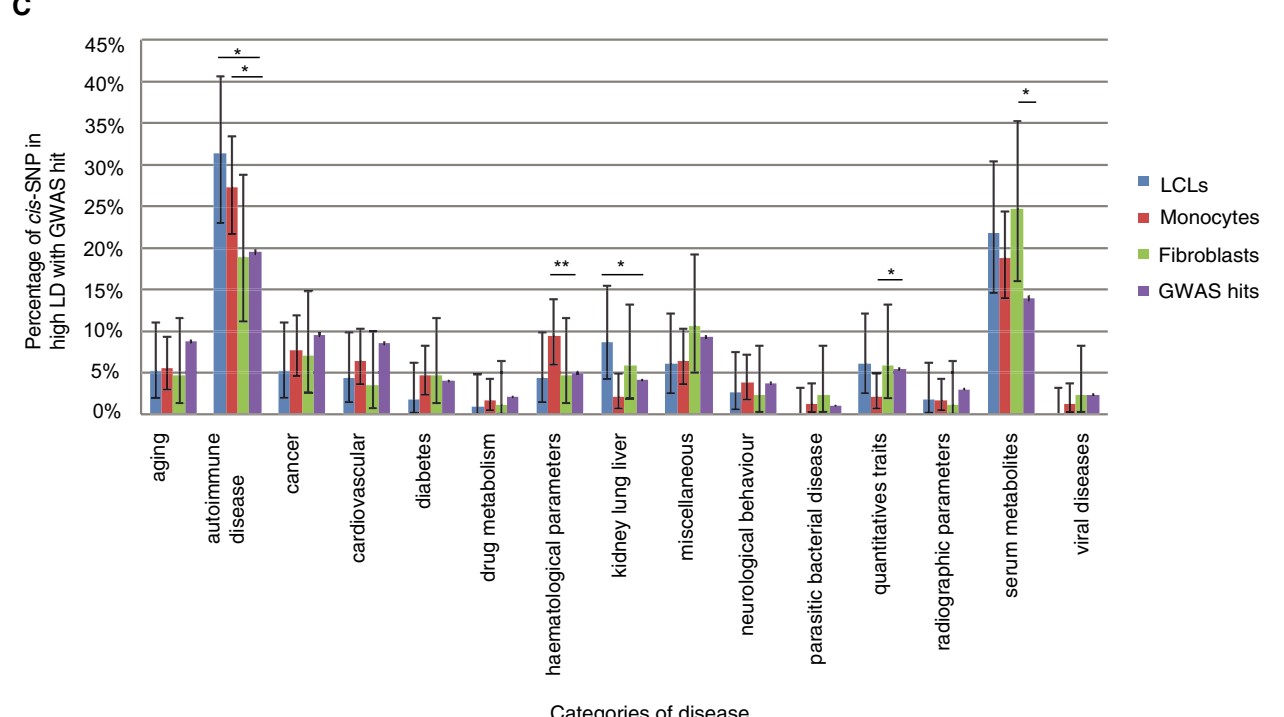

**Figure 2. Cis-rSNPs are linked with disease variants.**

A   The most significant *cis*-regulatory variant of *RP11-221J22.1* mapped by regression analysis in the monocyte population, *rs771767* ($P = 1.96 \times 10^{-6}$), is also a GWAS hit linked to multiple sclerosis.

B   Screenshot of the *rs771767* genomic region from the UCSC genome browser.

C   Proportion of *cis*-rSNPs in high LD with GWAS hits in each disease category. Both LCLs and monocytes showed significant enrichment in autoimmune diseases ($P < 0.05$). Monocyte-specific *cis*-rSNPs are enriched in hematological traits ($P < 0.01$, $*P < 0.05$, $**P < 0.01$).

*et al*, 1995; Schoenherr & Anderson, 1995) with an extensively documented repressor activity, *HFH4*, a FOX factor (Hoggatt *et al*, 2000; Myatt & Lam, 2007), or *PAX5* (Fazio *et al*, 2008). Unexpectedly, we also identified many "*de novo*" motifs among the most significant hits: 79% of the "*de novo*" motifs are bound by factors with repressive function (Fig 4A). To exclude bias due to the way in which matrices were produced with the "DNase I foot-printing" method, we examined footprint-generated motifs which

had not been called as "*de novo*" due to redundancy with other datasets (Neph *et al*, 2012). We observed a majority of "activators" (62%), a proportion more similar to the motifs from the TRANSFAC database (90%), suggesting no bias in the footprint-derived motif generation (Fig 4A). To further validate these findings, we assessed chromatin state at *cis*-rSNP sites where motifs are disrupted (see Methods) (Degner *et al*, 2012). We used in-house, high-depth H3K4me3 ChIP-seq data generated in LCLs. When looking at all

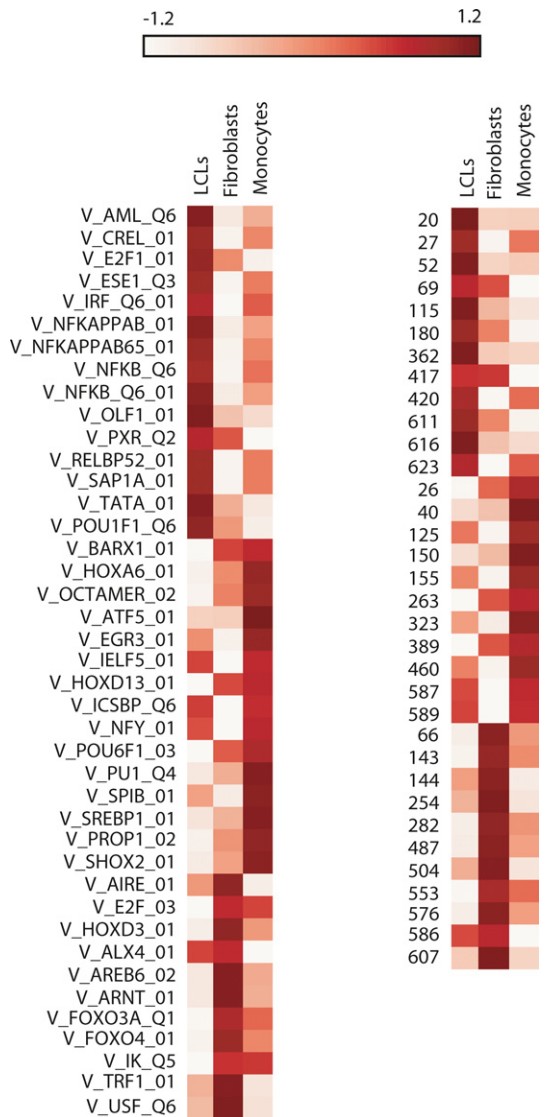

**Figure 3.    Heat map of differential motif disruption by *cis*-rSNPs according to the cell type.**

Heat map representing disruption frequency of motifs for TF binding sites (TRANSFAC, left panel; footprint-derived *de novo*, right panel) in LCLs, fibroblasts, and monocytes. Only motifs significantly enriched or depleted in at least one tissue are represented (*P*-value chi-squared < 0.05). Low disruption: white; high disruption: dark red. Numbers on the right panel correspond to the unique *de novo* motifs numbered 1–683 as defined by Neph *et al* (see main text).

*cis*-rSNPs  ($P < 1.10\text{E}^{-9}$),  we  observed  significant  enrichment (~1.2-fold, *t*-test,  $P = 5.4 \times 10^{-7}$) on the same haplotype of the expected overexpressed allele, confirming global enhancer activity for most of the *cis*-variants. In contrast, when we restricted the analysis to *cis*-rSNPs leading to the disruption of a "repressor" site, we observed the opposite trend (~1.2-fold, *t*-test, $P = 3.2 \times 10^{-4}$, Fig 4B). Overall, these data suggest that the disruption of repressor activity is an important source of heritable *cis*-regulatory variation and that repressor–DNA interactions are under-represented among annotated transcription factor binding sites (TFBS).

Next, we asked whether the *cis*-rSNPs associated with repressor activity were enriched in GWAS hits. We first identified 129 loci

**Table 3.    Table of TF biding sites with the most significant global activator or repressor activity.**

| Activity | Motif | *P*-value[a] | Bias=#over/#under[b] |
|---|---|---|---|
| Activators | V_PU1_Q4 | 1.33E-06 | 1.72 |
| | V_PU1_Q6 | 2.53E-06 | 2.61 |
| | V_CEBP_C | 1.11E-05 | 3.11 |
| | V_ELK1_01 | 2.04E-04 | 2.02 |
| | V_NFKAPPAB_01 | 2.66E-04 | 2.30 |
| | V_GADP_01 | 3.42E-04 | 1.78 |
| | V_P53_01 | 5.35E-04 | 2.28 |
| | V_GABP_B | 8.57E-04 | 1.61 |
| | 572 | 8.78E-04 | 1.44 |
| | 413 | 9.08E-04 | 1.60 |
| | 154 | 9.66E-04 | 1.74 |
| | V_NKX3A_01 | 1.07E-03 | 3.33 |
| Repressors | 421 | 1.46E-05 | 0.55 |
| | 481 | 6.43E-04 | 0.42 |
| | 275 | 9.12E-04 | 0.63 |
| | V_HFH4_01 | 1.05E-03 | 0.45 |
| | V_NUR77_Q5 | 1.40E-03 | 0.36 |
| | V_PAX5_01 | 2.22E-03 | 0.33 |
| | 373 | 2.46E-03 | 0.36 |
| | V_NFY_Q6_01 | 2.67E-03 | 0.43 |
| | 217 | 2.78E-03 | 0.65 |
| | V_NRSF_01 | 2.83E-03 | 0.58 |

[a]Based on global higher matrix score deviation toward the haplotype of the expected over or under-expressed allele.
[b]Total number of motifs with higher matrix score on the expected over or under-expressed allele.

where *cis*-variants are in high LD ($r^2 \geq 0.9$) with a disease hit and alter matrix scores for motifs associated to enhancer or repressor activity as described above. Among them, 66 *cis*-variants sit on a predicted repressor binding site. For example, *rs2303369*, which has been linked to age at onset of menopause, is in high LD ($r^2 = 0.9$) with *rs780100*, a *cis*-regulatory variant which disrupts a repressor binding site for the *de novo* motif "607", and is associated to differential AE of *NRBP1*, a gene with a growth-promoting role, in the monocyte population ($P = 2.7 \times 10^{-27}$) (Fig 4C and D) (Ruiz *et al*, 2012; Stolk *et al*, 2012). In order to validate the binding of a repressor factor, we examined allelic ChIP-seq signals for H3K4me3 at *rs780100* across all individuals that are heterozygous for this variant. We observed a significantly higher signal (~twofold, chi-squared test, $P < 0.001$ for all individuals except for MNC491 with $P < 0.05$) for active chromatin on the same haplotype as the less expressed allele, reinforcing our hypothesis of a repressor binding *NRBP1* promoter region (Fig 4E) (Light *et al*, 2014). It is also interesting to note that the risk allele is associated to the repression of this gene.

Taken together, the fine-mapped *cis*-rSNPs point toward frequent involvement of both known and currently uncharacterized transcription factors in the variation of gene expression in cell populations and in the pathogenesis of complex diseases. The global

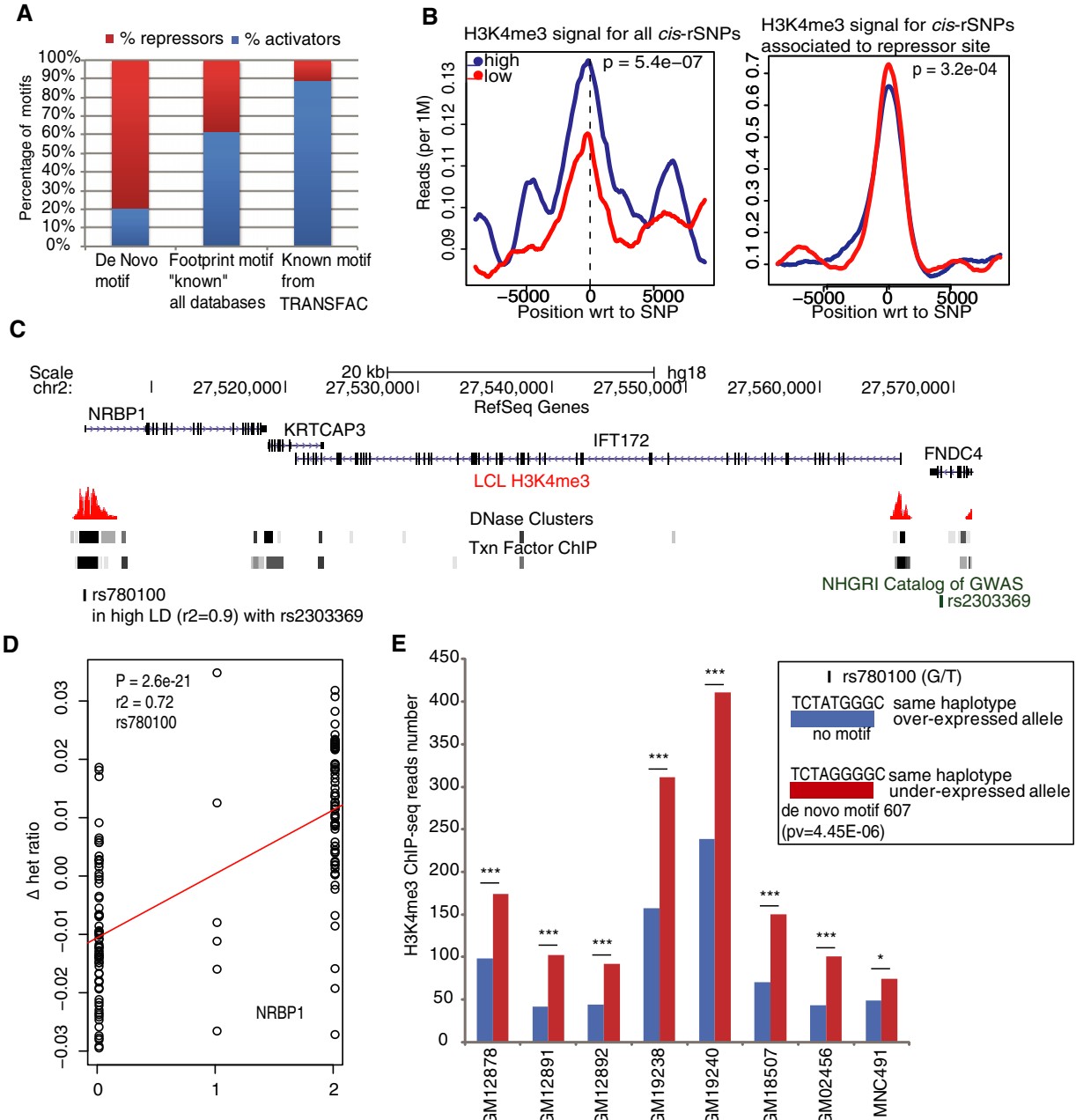

**Figure 4.  Footprint-based *de novo* motifs are enriched for repressor activity and linked to diseases.**

A    Proportion of disrupted motifs by *cis*-rSNPs for TF binding sites with globally significant activator or repressor activity (*P* binomial test < 0.01). Motifs are split into three categories: (i) footprint-based *de novo* motif with no match in any other major motif database; (ii) footprint-based *de novo* motif with a match in other major motif database: TRANSFAC, JASPAR, and UniPROBE databases (Neph *et al*, 2012); (iii) TRANSFAC motifs. Whereas TRANSFAC and previously known footprint-based motifs are mostly bound by activators, true *de novo* motifs showed enrichment for repressor binding sites.

B    ChIP-seq read depth is correlated to genotype of *cis*-rSNPs. Average normalized read depth according to genotype and across all LCL samples (*n* = 7) is depicted for H3K4me3 at all *cis*-rSNPs ($P < 1.1 \times 10^{-9}$) or at top *cis*-rSNPs associated to a change in a recognition motif for a repressor motif. For all mapped *cis*-rSNPs, ChIP-seq coverage is significantly higher ($P = 5.4 \times 10^{-7}$) for homozygotes for the expected overexpressed allele (blue line) as compared to homozygotes for the opposite allele (red line). We observed the opposite trend for repressor sites ($P = 3.2 \times 10^{-4}$).

C–E  Example of a repressor site in high LD with a GWAS hit. (C) Variant *rs2303369* is associated to age of onset of menopause. It is in high LD with *rs780100*, located in active chromatin and associated to differential AE of *NRBP1*. (D) AE linear regression graph. Regression test was carried out using phased AE data: *y*-axis shows the Δ het ratio, left dots correspond to heterozygotes carrying the B-allele in phased chromosome 2 ("0" on the *x*-axis), middle dots to homozygotes ("1" on the *x*-axis), and right dots to heterozygotes carrying the A-allele in phased chromosome 2 ("2" on the *x*-axis). (E) A recognition site for the *de novo* motif "607" is found at the location of *rs780100* on the same haplotype as the under-expressed allele; no motif is recognized on the other haplotype. Allelic ChIP-seq for H3K4me3 at this site showed significant bias toward the expected, under-expressed allele across all heterozygous individuals (*\*P* < 0.05, \*\*\**P* < 0.001) (first 6 samples = LCLs, last two samples are from fibroblast and monocyte (MNC491) cell population, respectively).

enrichments alone, however, are insufficient to confirm the role of specific TFs at defined loci.

### Genome-wide validation of NFκB allelic regulation in LCLs

For confirmation of TF–*cis*-rSNP interactions in a functional context, we developed a method that perturbs TF followed by monitoring genome-wide allelic expression measurements in living cells. As a model for this novel approach, we chose the factor NFκB. Activation of NFκB is known to regulate the expression of genes that are involved in the pathogenesis of inflammatory pathologies (Kempe *et al*, 2005; Sehnert *et al*, 2013) and SNPs associated to diseases are enriched in NFκB binding regions (Karczewski *et al*, 2013). Three motifs for NFκB binding ("V_NFKB_C", "V_NFKB_Q6_01", and "V_NFKAPPAB_01") are associated with significant global enhancer activity (*P* binomial test < 0.01) in this dataset (Supplementary Table S8). We observed that *cis*-rSNPs from 126 loci disrupted one of these NFκB binding sites are preferentially located in NFκB ChIP-seq peaks (ENCODE) (6.5-fold, chi-squared test, $P = 2.6 \times 10^{-23}$), when compared to other mapped loci in LCLs suggesting true regulation by this TF. Among these *cis*-rSNPs, 12 are in high LD with GWAS hits for 12 diseases and linked to differential allelic expression of 15 genes. Overall, both the literature and our own data support NFκB as an interesting biological model to test this novel approach. Briefly, we performed TNF-α induction coupled to inhibition of NFκB in LCLs followed by AE analysis on Illumina Human-Omni5-Quad BeadChips (see Methods). Samples used included two HapMap trios: one from the CEU (GM12891, GM12892, and GM12878) and one from the YRI (GM19239, GM19238, and GM19240) population. Validation of NF-κB knockdown was done using RT–PCR of known gene targets for NFκB (Supplementary Fig S8) (Mori & Prager, 1996; Catz & Johnson, 2001; Kang *et al*, 2007; Son *et al*, 2008). We looked for transcripts with AE differences in cells induced by TNF-α compared to cells induced by TNF-α in combination with inhibition of NFκB. We observed perturbation of AE of many known NFκB targets, such as *CXCL17*, *SERPINE2,* and IL-1A/IL-1B (Hiscott *et al*, 1993; Mori & Prager, 1996; Suzuki *et al*, 2006; Takegawa *et al*, 2008; Supplementary Table S9). Within the transcripts included for mapping in the CEU and YRI population, we identified 489 transcripts that are allelically regulated by NFκB, according to AE pertubation in at least two individuals heterozygous for the top *cis*-rSNP (see Methods). Using ENCODE NFκB ChIP-seq experiments, we observed significantly (~1.1-fold, chi-squared test, $P = 4 \times 10^{-17}$) higher NFκB signals at *cis*-rSNP locations from perturbed than non-perturbed transcripts, validating the global *cis*-regulation of perturbed genes by NFκB (Fig 5A). We then investigated whether the loci responding to NFκB perturbation and mapped in at least 1 cell population were associated to complex diseases according to an overlap between *cis*-rSNPs and GWAS hits (LD ≥ 0.9). We identified 26 transcripts that are perturbed by NFκB and had a regulatory variant linked to a complex disease, including immune-related and/or autoimmune diseases such as systemic lupus erythematosus (SLE), multiple sclerosis and Kawasaki disease (Supplementary Table S9). We focused on the *BLK* region, which is linked to SLE susceptibility, and for which we had previously fine-mapped the promoter region (Ge *et al*, 2009). Using significantly more individuals (118 CEU and YRI LCLs versus 53 CEU LCLs), we identified *rs998683* (RegDB = 1f) as the most strongly associated

SNP to differential AE of *BLK* (ENST00000259089.4) in the LCL population. This variant is located in the first intron of *BLK* gene and may act as an enhancer. We observed a change in AE of *BLK* after NFκB perturbation (Fig 5B–C) and an overlap of *rs998683* with an NFκB ChIP-seq peak, which strongly supports the role of NFκB in the regulation of BLK expression. Another example of converging functional data supporting the role of NFκB in the circulating plasminogen activator inhibitor-1 (PAI-1) concentration through the allelic regulation of *SERPINE1* is depicted in Supplementary Fig S9. Interestingly, we observed among the mapped and perturbed genes a significant enrichment of lincRNAs and processed transcripts (~twofold, chi-squared test, $P = 4.4 \times 10^{-20}$).

These results demonstrate that whole-genome perturbation of TF activity associated with allele-specific assessment and mapping can be successfully used to identify pathways and functional roles of regulatory variation associated to disease.

## Discussion

In this study, we mapped *cis*-regulatory variants in three cell types, examined their global impact on TF binding sites and demonstrated a new approach to validate the role of key regulators by combining allelic expression mapping data with targeted approaches to perturb TFs in living cells. The large fraction of discovered *cis*-regulatory variants in three cell types is in line with data from recent eQTL studies using RNA sequencing of blood cells or lymphoblasts (Lappalainen *et al* 2013, Battle *et al* 2014). We were able to achieve an equivalent number of associations using lower sample sizes due to the greater power of allelic expression measurements (Almlof *et al*, 2012). When shared associations are observed, the feasibility to fine-map *cis*-regulatory variants is improved by parallel application of meta-analyses across cell types as shown by their greater power to predict functionality. Finally, our results detect that up to 40–60% of *cis*-rSNPs that were originally mapped in each cell type show strong evidence of tissue independence, demonstrating a large pool of regulatory elements where sequence context predominates as determinate of variance. This reflects the strong sequence dependence of *cis*- regulation (Wilson *et al*, 2008).

Global investigation of the disruption of TF binding sites by *cis*-rSNPs can contribute to identifying important cell-type-specific regulatory factors and to distinguish functional variants associated to disease. Our analysis revealed that TFs with an inhibitory activity are likely to be more prevalent than previously thought. This inhibitory activity is more common for TF binding to *de novo* footprint-derived motifs than to known motifs from TRANSFAC. The fact that a TF can act as a repressor or an activator, depending on the chromatin or cellular context, appears to be a limitation for comprehensive mapping of repressor activity. Here, we focused on factors with a major global repressive activity. The knowledge of GWAS-associated variants implicated in repressor binding could be of practical importance in, for example, confirming the desired modulation of gene targets or their regulators in GWAS-based drug repositioning (Sanseau *et al*, 2012).

The intersection of *cis*-rSNPs with variants from the GWAS catalog reveals a large number of shared hits, with a remarkable enrichment of mapped variants from relevant cell types with disease types.

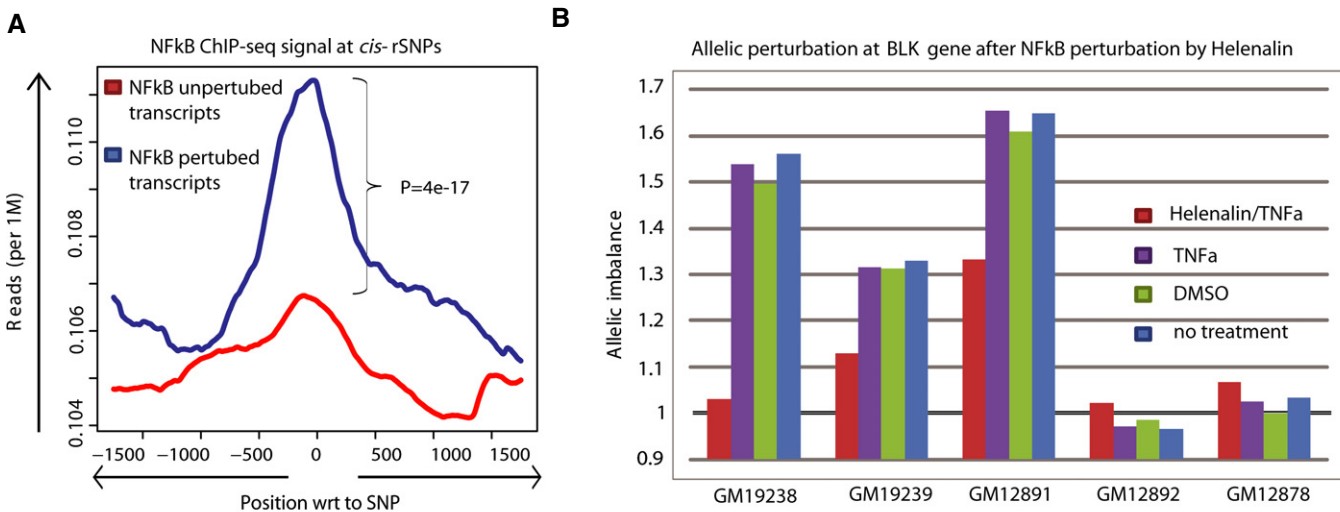

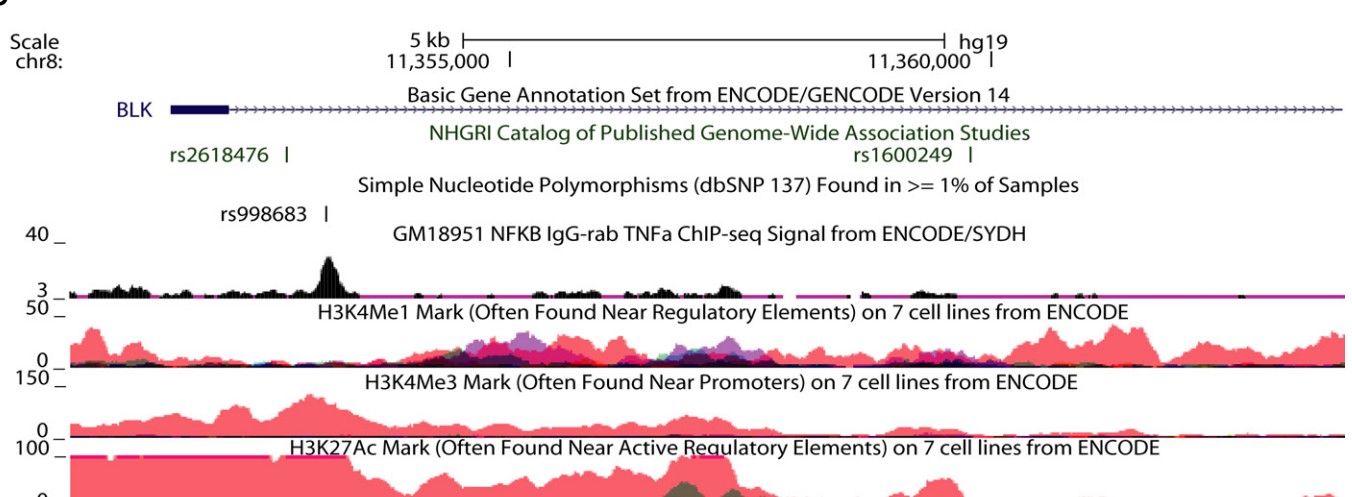

**Figure 5.  Direct assessment of allele-specific NFκB binding highlights target genes and its implication in diseases.**

A   ENCODE NFκB ChIP-seq signal at *cis*-rSNPs. We observe higher signal for transcripts that are allelically perturbed by NFκB. Lines represent normalized NFκB ChIP-seq read count for NFκB perturbed (blue) and unperturbed (red) transcripts. *P*-values were calculated with the chi-squared test using the read counts at the SNP position.

B   Allelic perturbation of *BLK* after treatment with helenalin/TNF-α, TNF-α, DMSO, or after no treatment. The value 1 on the *y*-axis represents equal expression of the two alleles of *BLK*. A complete or partial loss of allelic imbalance is observed after specific inhibition of NFκB (red) in cells heterozygous for *rs998683* (GM19238, GM19239, and GM12891). No difference in allelic expression between treatments is observed for homozygous individuals (GM12892 and GM12878).

C   Screenshot of the *rs998683* (*cis*-rSNP of *BLK*) region from the UCSC genome browser. An overlap between *rs998683* and the NFκB ChIP-seq peak is seen in a LCL sample.

Immune-related diseases are overrepresented in LCLs and monocyte-specific-associated SNPs, which are cell types with a well-known role in auto-immune diseases (Kwan *et al*, 2008; Zhang *et al*, 2008; Montgomery *et al*, 2010; Pickrell *et al*, 2010; Fairfax *et al*, 2012), reiterating the importance of using cellular lineages that match the disease biology. We observe that phenotypically important SNPs are enriched for complex transcriptional variation in populations where multiple independent transcripts are affected. This intriguing observation in some loci may have a biological basis, but in practice, it suggests that comprehensive characterization of genetically variable transcription in the vicinity of trait SNPs is essential to assign functional mechanisms to disease haplotypes (Verlaan *et al*, 2009).

A recent method proposed to investigate endogenous regulatory elements by selectively altering their chromatin state using programmable reagents (Mendenhall *et al*, 2013). Although promising, this method does not allow for genome-wide assessment of TF regulatory activity and their associated transcripts. We demonstrated a new approach for *in vivo* exploration of genome-wide functional activity of a transcription factor, focusing on NFκB given its key role in immune regulation. We validated allele-specific differences in TF binding in the human genome and successfully identified transcripts that are allelically regulated by NFκB in LCLs and whose regulatory activity is associated to complex diseases. We noticed an enrichment of lincRNAs and processed transcripts in genes regulated by

NFκB. We hypothesize that lincRNAs are more easily perturbed due to their lower stability versus protein-coding genes or that there are fewer post-transcriptional mechanisms that buffer the effect of *cis*-rSNPs in lincRNAs. This result also supports a potential role of non-coding RNA in modulation of inflammatory processes. Our novel approach to perturb NFκB and monitor the consequences of the perturbation on a genome-wide scale can be generically extended to other transcription factors or combinations of them in different cell types. Moreover, this approach may be used to test specific TF activity on isolated cells from patients and thereby identify allele-specific differences with controls.

Future analyses using allele-specific expression data for mapping *cis*-variants and for *in vivo* genome-wide assessment of TF activity in populations of diverse cell types hold tremendous promise for the large-scale identification of the specific causal variants that affect gene expression detected by genome-wide association studies for an assortment of complex diseases.

# Materials and Methods

## Cell preparation and cDNA synthesis

All LCLs were obtained from Coriell (Camden, NJ, USA) and cultured as previously described (Ge *et al*, 2009). Fibroblast cell lines were obtained from Coriell and the McGill Cellbank (Montreal, QC, Canada) and grown in medium containing a-MEM (Sigma-Aldrich, Oakville, ON, Canada) supplemented with 2 mmol/l L-glutamine, 100 U/ml penicillin, 100 mg/ml streptomycin, and 10% fetal bovine serum (Sigma-Aldrich) at 37°C with 5% $CO_2$. At 70–80% confluence, the cells were harvested and stored at −70°C until RNA and DNA extraction. RNA was extracted from cell lysates, and we applied a cDNA synthesis protocol as previously described (Ge *et al*, 2009). Circulating monocytes were collected from healthy adult blood donors of European origin ($n = 188$) recruited from the United Kingdom National Blood Service Centre in Cambridge, UK, as part of the Cardiogenics Transcriptomic Study (Garnier *et al*, 2013). The Cambridgeshire 1 Research Ethics Committee approved the donor recruitment. CD14[+] magnetic microbeads (autoMACS Pro, Miltenyi Biotec, Bergisch Gladbach, Germany) were used to isolate monocytes from whole blood. RNA was extracted from cell pellets, and cDNA was prepared as previously described (Almlof *et al*, 2012).

## RNA-seq samples preparation

Total RNA from LCLs (x3), fibroblasts (x4), and primary monocytes (x8) were extracted from cell lysates using the miRNeasy Mini Kit (Qiagen, Mississauga, Canada) (including a DNase I treatment step) with the quality assessed by Agilent 2100 BioAnalyzer (Agilent Technologies, Palo Alto, CA, USA). Libraries for RNA sequencing were prepared according to the Illumina TruSeq protocol. The quality of each library was assessed by Agilent 2100 BioAnalyzer. Samples were indexed and sequenced on Illumina Genome Analyzer II (paired-end 2 × 76 bp) or on Illumina HiSeq 2000 (paired-end 2 × 100 bp). High-quality RNA-seq reads were aligned to the human reference genome build hg19 using Tophat v1.4.1 (Trapnell *et al*, 2012). Annotated transcripts (Gencode V15) with median FPKM score > 0.01 across samples were considered as expressed (Trapnell *et al*, 2010).

## AE mapping and normalization of allele ratios in Illumina Beadchips

AE mapping was performed as previously described (Ge *et al*, 2009) except for the signal intensity normalization at heterozygous sites, which followed a slightly modified approach (Grundberg *et al*, 2011). Briefly, approximately 200 ng of genomic DNA and a 50–300 ng double-stranded cDNA sample was used for the parallel genotyping and AE analysis on the Illumina Infinium Human1M or Human1M-Duo SNP bead microarray according to the manufacturer's instructions. Raw data were processed in the genotyping module (Ver. 3.3.7) of BeadStudio software (3.1.3.0), filtered and normalized. For transcript-based AE regression tests, associations were carried out using average intensity signal for SNPs (minimum three per transcript) across any annotation from GENCODE version 15. We used 1000 Genomes project data as a reference set (release 1000G Phase I v3, updated 26 Aug 2012) for the imputation of genotypes from our panel of HapMap individuals. Untyped markers were inferred using algorithms implemented in IMPUTE2 (Howie *et al*, 2009). $R^2$ was used as an imputation quality control metric, which estimates the squared correlation between imputed and true genotypes. We systematically removed all poorly imputed markers with $r^2 < 0.8$. For each gene, only the most highly expressed isoform with minimum FPKM value of 0.01 was retained. In regression tests, we decided to exclude intensity signal from genomic regions overlapped by more than one gene to avoid conflicting data. We called "partial-length transcripts" in cases where only part of the AE data were used due to these low confidence regions. For "full-length" transcripts, no region was excluded. The AE associations were tested in phased chromosomes with Δ het ratio data correlated with local (±500-kb flanking sequence) genotypes, the marginal (at 0.01 permutation significance level) associations observed in either population from non-overlapping transcripts. The number of tested SNPs per cell population can be found in Supplementary Table S1. Overall, 4939556, 7659025, 4947257 and 4933245 SNPs with MAF ≥ 0.05 were tested in CEU LCLs, YRI LCLs, fibroblasts, and monocyte population, respectively, using 12411 full- and 3901 partial-length transcripts. To be conservative, loci were included only if the *P*-value of the most significant association was less than the *P*-value at 1% FDR, which was $3.282 \times 10^{-6}$, $1.263 \times 10^{-6}$, $4.238 \times 10^{-6}$ and $3.276 \times 10^{-5}$ for the CEU LCLs, YRI LCLs, fibroblasts, and monocytes, respectively.

## Shared associations and Fisher combined test

For each locus, all primary associations from each population were compared to the first percentile of mapped SNPs in others cell population. A *P*-value cutoff of $1.1 \times 10^{-4}$ was used in the secondary population. At this threshold, > 95% of mapped associations showed the same direction of effect between two populations. When at least one regulatory variant was found in common, we considered this locus as sharing regulatory activity between these populations. A Fisher's combined approach was then used to improve mapping resolution and generate new associations for shared loci across populations.

To allow comparison of our approach to recent works reporting higher level of sharing, we used same approach for comparing *P*-values between datasets. Specifically, we took lead association in one tissue (based on our definition by FDR) and fetch exactly the

data for same SNP-transcript pair in the other tissues. We used as input these *P*-value lists in the R package "qvalue" and run the default setting in the two datasets separately. The output $\pi 0$ in the summary file is used to calculate $\pi 1$ values $(1-\pi 0)$, which represents the proportion of shared hits among the tested. This approach is more liberal and allows for uncertainties of mapping accuracy in differently powered datasets; consequently, pairwise sharing estimated based on this approach is substantially higher (40–60%).

To evaluate the potential functionality of these new associations, we intersected our data with the RegulomeDB (RegDB) database (Boyle *et al*, 2012). Categories 1–5 were used to indicate active regulatory regions (a lower score indicating stronger evidence of functionality), whereas categories 6 and 7 are attributed to SNP with low potential of functionality.

### ChIP-seq samples preparation

Cells were cross-linked with 1% formaldehyde at room temperature for 10 min. After quenching with glycine for 5 min (125 mM glycine per ml of media), the cells were washed twice with ice-cold PBS. Cells were collected after each wash by centrifugation at 2,000 *g* for 5 min. Cell pellets were flash-frozen and stored at −80°C. Frozen pellets were thawed, and cells were lysed in Farnham lysis buffer (5 mM PIPES pH 8.0, 85 mM KCl, 0.5% NP-40 and protease inhibitors) for 10 min on ice. After centrifugation and a wash with 1 ml of RIPA buffer containing 50 mM Tris–HCl pH 8, 150 mM NaCl, 1% NP-40, 0.5% sodium deoxycholate, 0.1% SDS, and protease inhibitors, lysates were diluted with 500 μl of RIPA buffer to proceed to the sonication step. Cells were sonicated in non-stick tubes under conditions optimized to yield soluble chromatin fragments in a size range of 100–250 base pairs. Chromatin from 40 million cells was sonicated for 10 min using a Branson 250 sonicator at 20% power amplitude (pulses: 10 s on and 30 s off). Lysate was clear by centrifuging at 12,000 *g* for 10 min at 4°C to eliminate cellular debris. Chromatin was then flash-frozen and stored at −80°C or used immediately for the next step. Before each immunoprecipitation, chromatin was pre-cleared with 50 μl of pre-washed ProteinA-magnetic beads (Invitrogen; 100-02D) to avoid non-specific binding. Immunoprecipitation was carried out for 12 hours by rotation at 4°C in 500 μl of chromatin/RIPA buffer supplemented with protease inhibitor cocktails (Roche; 04 693 159 001) and PMSF. We used 10–30 million cells and 2–5 μg of the following antibodies for each assay: H3K4me1 (abcam; ab8895) and H3K4me3 (Diagenode; #pAb-003-050). After overnight incubation, samples were rotated with 100 μl of pre-washed Protein-A-magnetic beads at 4°C for 1 h. The beads were then collected by brief centrifugation at 2,000 *g* following by the use of a magnetic rack. Beads were washed five times with 1 ml of LiCl wash buffer (100 mM Tris pH 7.5, 500 mM LiCl, 1% NP-40, 1% sodium deoxycholate) by resuspending the beads and keeping them on ice for 10 min. Bound chromatin was then eluted from the beads by incubation with 200 μl of elution buffer (50 mM Tris–HCl, pH 8.0, 10 mM EDTA, 1.0% SDS) at 65°C for 1 h with vortexes performed every 15 min. This was followed by a centrifugation at 14,000 *g* at room temperature for 3 min. The eluted chromatin and the "input" sample were then incubated at 65°C overnight after adding 0.2 M of NaCl to reverse cross-links. Samples were then treated with RNase A at 37°C for 30 min, followed by digestion with proteinase K at 55°C for 1 h. Immunoprecipitated DNA was then purified using QIAquick PCR

Purification Kit (QIAGEN; 28104) and eluted to a final quantity of 30 μl. Enrichments of interesting regions were validated using real-time PCR experiments. Primers were designed to genomic sites known to bind H3K4me1 and H3K4me3 enriched or not enriched (negative control) regions. Library preparation for ChIP-seq assays was carry out using Paired-End DNA Sample Prep Kit V1 (Illumina; PE-102-1001) and sequenced using the Illumina Genome Analyzer II (2 × 76 bp) or HiSeq Sequencing System (2 × 100 bp). The panel we used consisted of 7 HapMap samples: 3 CEU LCLs (GM12891, GM12892, and GM12878) and 4 YRI LCLs (GM19238, GM19239, GM19240, and GM18507); 2 fibroblast cell lines from Coriell: GM02456 and GM2555; 2 purified monocyte samples: MNC491 and MNC492. Reads were trimmed for quality (phred33 ≥ 30) and length ($n \geq 32$) using Trimmomatic v. 0.22 (Bolger *et al*, 2014). The filtered reads were aligned to the hg19 reference genome using BWA v. 0.61. Peaks were calls using MACS v. 1.4.2 (Feng *et al*, 2012).

### Motif over-representation and allelic positional bias

Matrices for TRANSFAC (version 2009.4) and *de novo* footprint-derived motifs (Neph *et al*, 2012) were used in association with the FIMO motif scanning software, version 4.9.0, using a $P < 1 \times 10^{-4}$ threshold, to find all motif instances ± 15 nucleotides from a mapped *cis*-rSNP sitting on a DHS footprint region (Grant *et al*, 2011). All motifs displaying no change in matrix affinity score, according to *cis*-rSNP genotypes, were discarded. To account for multiple testing, we used a Bonferroni correction (0.05/1380 = 3.62e-05). Among the *de novo* motifs, 58% matched matrices from other databases (TRANS-FAC, JASPAR or UniPROBE). These motifs were not considered as *de novo* for the analyses. After normalization to a mean value of 0 and variance 1, a heat map with 1 row per motif instance was generated using matrix2png (Pavlidis & Noble, 2003), version 1.2.1. The full dataset is accessible in Supplementary Table S6.

### GWAS intersection

The GWAS catalog was obtained from http://www.genome.gov/admin/gwascatalog.txt on June 26, 2012. We grouped SNPs into classes of similar diseases or traits according to the classification used by Maurano *et al* (2012).

### Perturbation of NFκB

LCLs from HapMap trios from the CEU (GM12891, GM12892, and GM12878) and YRI (GM19239, GM19238, and GM19240) LCL populations were used. Cells were plated in 6-well plates with 500,000 cells/ml in 2 ml one day prior to the experiment. Cells were either directly treated with TNF-α (3 ng/ul) for NFκB activation or primarily transfected for one hour with helenalin (5 μM) (EMD Chemicals, USA) in order to inhibit the activation of NFκB (p65) (Lyss *et al*, 1998). Helenalin is a sesquiterpene lactone that acts as a specific NFκB DNA binding inhibitor by irreversibly alkylating free sulfhydryls of the cysteine residues on the p65 subunit. Following this inhibition, cells were stimulated with TNF-α (3 ng/μl) at time points consisting of 4, 6, 8, 12, 24, and 48 h in order to select for the ideal stop point. Validation of the perturbation of NFκB and induction by TNF-α was done by RT–PCR for genes targeted by NFκB including IL-6, IL-8, IL-1a, and Bcl-2. The most optimal time point to stop the

experiment was deemed 8 hours post-transfection. Total RNA and DNA were extracted for dscDNA synthesis. Differential AE was assessed on Illumina HumanOmni5-Quad BeadChips for each experimental condition: (1) inhibition with helenalin followed 1 h later by activation by TNF-α (H-TNF-α); (2) activation by TNF-α (TNF-α); (3) DMSO; and (4) no treatment. We selected NFκB perturbed genes using the following criteria: differential allelic expression must be higher in (2) than in (1) and with more than 1.2-fold change between H-TNF-α and TNF-α in at least two individuals. We required that this variation was measured in at least three independent heterozygous SNPs in both conditions.

### RT–PCR

Total RNA was annealed to 500 ng of random primers. First-strand cDNA synthesis was performed using SuperScriptII reverse transcriptase (Invitrogen Corporation, Carlsbad, CA, USA) according to the manufacturer's recommendations and as described above. The cycling conditions on the Rotor-Gene™ 6000 real-time rotary analyzer were 4 min at 95°C, 40 cycles ×20 s at 95°C, 30 s at 58°C and 30 s at 72°C, followed by the dissociation protocol at 72°C. Results were analyzed using the comparative CT method. The CT mean and standard deviation of each technical replicate were calculated, and the mean CT values were then normalized to the 18S mean CT value. Primers were designed using the Primer3 v. 0.4.0 software (http://frodo.wi.mit.edu/), and all primer sequences used can be found in Supplementary Table S10.

### Data access

Primary Data:
cDNA and gDNA raw data for YRI LCLs can be accessed through the GEO accession number GSE52442.
RNA-seq data for primary monocytes and YRI LCLs can be accessed through the GEO accession number GSE53837.
ChIP-seq data for primary monocytes can be accessed through the GEO accession number GSE53837.
cDNA and gDNA raw data after NFκB perturbation (treatment and controls) for the two LCLs trio can be accessed through the GEO accession number GSE61254.
Referenced Data:
cDNA and gDNA raw data for monocytes and fibroblasts can be accessed through accession number EGAS00000000119 at EGA and the GEO accession number GSE52442, respectively.
RNA-seq data for fibroblasts can be accessed through the GEO accession number GSE53837.
ChIP-seq for fibroblasts and LCLs can be accessed through the GEO accession number GSE53837.

**Supplementary information** for this article is available online:
http://msb.embopress.org

## Acknowledgements

The authors are grateful to all of the study participants, including the volunteers from the Cambridge BioResource who donated blood for this study (http://www.cambridgebioresource.org.uk/). SNP genotyping for AE determination in the monocyte samples was performed by the SNP&SEQ Technology Platform in Uppsala, Sweden (www.genotyping.se), and National Genomics Infrastructure, Science for Life Laboratory, Sweden. We thank Anders Lundmark for his assistance. This study was supported by the Cardiogenics EU FP6 project (Grant Number LSHM-CT-2006-037593), the European Sequencing and Genotyping Infrastructure (ESGI) (EU FP7 2007-2013 under Grant Number 262055), the Swedish Research Council (Grants No A028001 and C0524801), the Knut and Alice Wallenberg Foundation (KAW 2011.0073), Genome Quebec, Genome Canada, FRSQ (RMGA), and a Canadian Institutes of Health Research (CIHR) grant awarded to TP. In addition, TP holds a Canada Research Chair.

## Author contributions

VA, AS, A-CS, and TP conceived the research and designed the experiments. LR supervised cell collection for ChIP-seq. AHG, FC, PD, WHO designed monocyte dataset collection. VA, AS, CW, and SC conducted the experiments. VA, NL, BG, and TP designed the computational and analytical methods. VA, AS, NL, JCA, PL, BG, TK, MC, and TP analyzed the data. VA and TP drafted the manuscript, and all authors contributed to the final manuscript writing and its revisions.

## Conflict of interest
The authors declare that they have no conflict of interest.

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
