## [Review Process File · Molecular Systems Biology]

Allelic expression mapping across cellular lineages to establish impact of non-coding SNPs

Veronique Adoue, Alicia Schiavi, Nicholas Light, Jonas Carlsson Almlöf, Per Lundmark, Bing Ge, Tony Kwan, Maxime Caron, Lars Rönnblom, Chuan Wang, Shu-Huang Chen, Alison H Goodall, Francois Cambien, Panos Deloukas, Willem H Ouwehand, Ann-Christine Syvänen, Tomi Pastinen

Corresponding author: Tomi Pastinen, McGill University

Review timeline:

Submission date:	13 January 2014
Editorial Decision:	13 March 2014
Revision received:	12 June 2014
Editorial Decision:	20 July 2014
Revision received:	18 August 2014
Editorial Decision:	31 August 2014
Revision received:	09 September 2014
Accepted:	11 September 2014

Transaction Report:

1st Editorial Decision

13 March 2014

Thank you again for submitting your work to Molecular Systems Biology. First of all, I would like to apologize for the delay in getting back to you. We have now heard back from the three referees who agreed to evaluate your manuscript. As you will see from the reports below, the referees find the topic of your study of potential interest. They raise, however, substantial concerns on your work, which, I am afraid to say, preclude its publication in its present form.

While reviewer #1 and #3 are rather positive, reviewer #2 is much less supportive. The major issue refers to the need of a global overhaul in the quality of the presentation and discussion of the results to improve the logic and the flow of the manuscript and, crucially, to document the analysis and methods in a much more rigorous manner. We agree with the assessment of the reviewers and think that an in-depth rewrite of text is necessary.

If you feel you can satisfactorily deal with these points and those listed by the referees, you may wish to submit a revised version of your manuscript. Please attach a covering letter giving details of the way in which you have handled each of the points raised by the referees. A revised manuscript will be once again subject to review and you probably understand that we can give you no guarantee at this stage that the eventual outcome will be favorable.

Reviewer #1:

This paper provides a good and quite comprehensive eQTL (allele specific) study using chip read out manner. They find a number of allele specific QTLs (ie, QTLs which change allele specificity) in 3 different cell types - LCLs, Monocytes and Fibroblasts. They then do motif analysis on this and specifically stimulate some LCLs with TNF alpha to help explore the impact of change in NFkB expression.

Overall I was impressed by this analysis and do not have major technical concerns.

I do however have some presentational concerns. I think readers not used to this data would appreciate a single data point example of the an AE QTL - rather like Figure 2C, but also showing when the read out SNP is relative to the QTL SNP in potentially a cartoon.

The more serious point is that it is clear that the authors struggled with the problem of both differential power and the inevitable point where the FDR cutoff means that a QTL might be present in many cell types but called in only 1 or a subset due to these issues. therefore the statement that most QTLs are shared is I think instinctively correct, but Figure 1A and Figure 1B visually read in the opposite direction. Given this I wonder if Figure 1A and 1B is really the way to present this information (though it does provide the numbers). I would encourage the authors to be much more up front about not just differential power in teh samples, but also the inherent problem of a continuum of effect sizes, and having to threshold each scenario. (Arguably one should use one of these new joint models that handles this statistically, such as from the Pritchard group, but I would not necessarily insist on this). Secondly I would question the Venn diagram presentation, perhaps focusing on the inverse - the number of QTLs with some clear cut single tissue specificity.

Pertinent to this is I am not sure one can be quite so strong about the link between overall tissue expression levels and QTL - in some sense this is about the natural effect size vs the noise - in other words, a QTL as a proportion of the transcription, might be present at equal levels in low and high expressed genes, but only detectable compared to noise variation in the higher expressed case. The authors should soften this claim.

Other points:

I would prefer a consistent ordering of how the cell lines are listed in the paper text and the figures. There is a variety of orders. I would suggest doing this in ascending sample size (ie, Monocytes last).

I could not work out what hte Y axis really was in Figure 6B. If it is like the other examples (eg, Figure 2C) why not use that? And label the axis.

There is a typo "GO accession number" should be "GEO accession number" I think.

Reviewer #2:

Adoue et al. use allelic expression to characterize various features of genetic regulatory variation, from noncoding transcripts to GWAS relevance and transcription factor binding. As it is often the case in these kinds of descriptive studies where the aim is to characterize functional variation, there is no single hypothesis or one key conclusion. Their conclusions - of the pervasiveness, tissue sharing, and GWAS relevance of cis-regulatory variation - are rather incremental without tremendous novelty value, but especially the NFKP experiments to understand the role of TF binding in gene regulation are interesting. The study relies mostly on established methodology and focuses on biological discovery instead of novel computational approaches.

The paper touches many topics, and I had a hard time seeing how some of the analyses link to each other. It seems that the authors tried to think every single analysis that they could do with these data, and report all of them - instead of describing a well-defined set of analysis to answer a well-defined question. This is illustrated by the last paragraph of Introduction that is essentially a list of various

topics that don't all have a lot to do with each other as biological phenomena, and this continues in Results that jumps from one thing to another. The authors should put substantial effort into building a proper narrative. Focusing on the TF binding part and maybe omitting some of the less novel analyses from the beginning of the paper (at least from the main text/figures) would make this paper more interesting to the readers.

Most of the conclusions that the authors present might be correct, but in the end I was unable to finish this review because the results and analysis are not described to a sufficient degree that I could really evaluate them. The sloppy reporting of the results is completely unacceptable. I got to page 9, wrote 2 pages of "minor comments", and stopped there. Reading the paper leaves me with a feeling that I hardly know what the analysis is, what the result is, and/or how significant it is. There are fold enrichments but no p-values, and vice versa. Several figures are lacking proper labeling. Results are frequently reported without any reference to the actual data and numbers. There are figures without references in the text. This is so frequent that I'm not even going to list all of them systematically (a pretty comprehensive list is below), and on page 9 I decided that this is beyond my job as a reviewer. I should not need to teach the authors - many of which have published quality work before - how to report data and results. They should have done this before even thinking of submitting this manuscript.

Please note that in the pull-down menu of different aspects of the manuscript, it was considerably difficult to say whether the quality of methods and conclusions was adequate or low - while my impression is that the methods and conclusions are probably OK, I cannot be sure of this because of the sloppy reporting, and thus I have no other option but to err on the cautious side.

Specific issues:

- When discussing lincRNAs, the authors need to consider a recent paper on the topic: PMID 24268656. Some of their results reported here as novel are already included in the Popadin et al. paper (as well as in Kumar et al. that the authors already refer to), and the results of that paper contradict some of the statements e.g. in the introduction (low rate of cis-regulation in lincRNAs is contrary to the Popadin et al. paper). In general, I'd suggest that the authors put substantially less emphasis in this section because few of the results are that novel, and it distracts the reader from the more important and interesting points.
- Abstract: I believe that "bind" should be "bound" in "...motifs, predominantly bind by repressive factors..."
- Introduction: "cis-regulatory mechanisms free of trans-acting variation and environmental effects" is a false statement. For example, the NFKP results in this paper are an elegant demonstration of how trans or environmental effects can modify cis-regulatory effects!
- Introduction: "Studies of AE-mapping in lymphoblastoid cell-lines (LCLs) have revealed that approximately 30% of all loci have significant AE imbalance, with cis-rSNPs explaining more than 50% of the population variance in AE". Reference? How is this estimated in population samples - if it's a total across many samples, one would eventually reach 100% by sampling deeper (analogously to the number of variants discovered in sequencing studies), and if measured in an individual samples, this number is sensitive to technology and e.g. RNA-seq depth as well as statistical definition of AE.
- The number of samples used in this study is mentioned late in results. This is key information that should be in the abstract and early in the description of the study.
- The authors need to specify in the methods and also briefly in the methods what exactly they are doing with introns and how they are assigning sites to transcripts (what are full and partial-length transcripts? What is "data from not conflicting regions"). This is a key point. If the authors are in any way annotating sites from the region of same gene into different transcripts and analyzing them independently, this is not appropriate. This is particularly relevant when reporting cis-rSNPs associating to several transcripts. The 35% of such SNPs seems like a high number to me, but since I don't know how transcripts are grouped, I can't really say if it makes sense or not.
- In the methods, I assume that Ge et al. is the reference, but the method needs to be described briefly anyway. One should not need to look into a supplement of another paper to find out the basics of the analysis.
- Results "Through the simulation analysis we observed that the percentage of missing causal SNPs was below 5%..." What simulations? I don't find a description of these anywhere, not in this manuscript or in Ge et al.
- All the barplots should have error bars

- Fig 1A: What are these numbers? "Associations" = number of transcripts, or something else?
- Fig 1B and reference to it in the text: where are the p-values? Also, it makes no sense to show H3K4me3 as bars and H3K4me1 as lines - they should both be bars, with different colors.
- Fig 1D: The weird 3D lines serve no other purpose than making it difficult to read the plot. Make it two-dimensional. Also, I think that the result shown in this figure has been seen so many times before that it could well be in the supplement.
- Fig S4: Cutting the y-axis at 40-49% makes the difference look much more dramatic than what it actually is. Make it 0-49%.
- "This result [Fig 2A] suggests that protein-coding and non-coding transcripts shared same cis-regulatory variants." As far as I understand, these results don't say anything about the variants being the same, only that they have similar properties.
- The proportion of sharing between populations seems very low, which disagrees with earlier papers. How do the authors explain this?
- "Among these cis-rSNPs representing 7% (n=24891) of all mapped cis-rSNPs, we observed a significant enrichment ($P=1 \times 10^{-9}$) of associations at intermediate distance (5kb to 200kb) from the TSS, when considering associations ranked 1 for each transcript only." Isn't this true almost by definition? I find it hard to imagine a scenario of multi-transcript associations where this isn't true - hence, this observation carries no biologically interesting information.
- "These observations argue for an impact of genetic variants on higher order chromatin function, as reported by us earlier for a single disease locus (Verlaan et al, 2009), which we now show to be a frequent phenomenon across human cell types." These results do not justify such a strong statement, and chromatin function is a very vague concept. The results here are suggestive at best without a more detailed analysis of the properties of the transcripts, and potentially also additional data of e.g. open chromatin in these regions.
- Is Fig S6 lacking legends or captions or something? What are these scores and bars of different colors?
- Fig S7 is lacking axis labels, and there are more bars than labels.
- Fig 3A is a table, not a figure.
- Fig 3C is lacking an axis label.
- "Non-coding genes are rarely suggested as candidates for disease". This is not true - plenty of examples of ncRNAs with a putative role in disease are known, and this is widely recognized by the community.
- Identification of cell-specific active motifs disrupted by cis-rSNPs : Where's the data? Fig 4a shows "selected" results - selected based on what? The full data needs to be provided.
- Fig 4A: what are these codes for the motifs? What do the numbers correspond to?
- "We also found that many de novo footprint-derived motifs are more specific to one of the three cell-types, suggesting selective activity for the associated transcription factors." Where's the data?
- The section about de novo motifs needs to begin with a brief summary of what these motifs are. A reference to Neph et al. is not sufficient, because the nature of these motifs is critical for understanding the validity of the results.
- "We identified 110 motifs with significant allelic bias" out of how many? Fig 4b is also a table, not a figure. It needs to include some quantification of the bias in addition to the p-value.
- "we looked only at motifs with a predicted regulatory activity with a p-value below 0.01." What's this p-value?
- I don't find any reference to Supplementary Figures 8 and 9.

Reviewer #3:

The authors present maps of allelic-expression (AE) associated SNPs using quantitative measurements of AE on Illumina Human1M BeadChips in different cell lines and populations, thereby extending work that has been published by Ge B et al. in *Nature Genetics* (2009) on the same platform. The authors further extend work on eQTLs published using different expression platforms (such as RNA-Seq, which has a different sensitivity towards detecting AE when compared to Illumina Human1M BeadChips). The authors use this platform to map common SNPs associated with cis regulation of both protein coding and non-coding genes ("cis-rSNPs").

There is a number of new observations described in this manuscript, facilitated by the sensitive parallel DNA/RNA genotyping approach used, that in my view will be of interest to MSB readers. The sensitivity of this platform enables the authors to comprehensively assess cis-rSNPs associated

with non-coding RNA expression, which includes a number of SNPs that previously had been identified in GWAS studies. Amongst these there is an abundance of cis-rSNPs with inferred repressive effects. Additionally, by perturbing NF B in lymphoblastoid cells the authors are able to identify a number of cis-regulated transcripts with altered AE upon perturbation, which shows the relevance of alerting/perturbing transcriptional networks for carrying out comprehensive AE analyses.

My major criticism is rather a stylistic, than a scientific one. The manuscript is currently on the lengthy side, with lots of redundancy, and would significantly benefit both from English language editing and from further streamlining. In some instances, colloquial expressions are being used by the authors where more scientific ones should be used for clarity (one example, on p. 5, the authors write that "protein-coding and non-coding transcripts shared same cis-regulatory variants", where I assume that what the authors meant to stress is the finding that the percentage of cis-regulated transcripts, as defined by their approach, was similar for those corresponding to non-coding vs. protein-coding transcripts).

One additional more specific scientific comment is whether the authors can quantify to what extent their finding described on p. 6, i.e. that only 0.4% of rSNP associations were >200kb apart from the target transcript, may be affected by the low sensitivity of the BeadChip platform towards detecting associations involving distant sites (i.e., both RNA and DNA can contribute to the AE mapping with this parallel DNA/RNA genotyping approach, which may have significantly contributed to the peaking in identified cis-rSNPs for genic (vs non-genic) regions, displayed in Figure 1D).

1st Revision - authors' response

12 June 2014

Point-by-point response to the referees:

Reviewer #1: *This paper provides a good and quite comprehensive eQTL (allele specific) study using chip read out manner. They find a number of allele specific QTLs (ie, QTLs which change allele specificity) in 3 different cell types - LCLs, Monocytes and Fibroblasts. They then do motif analysis on this and specifically stimulate some LCLs with TNF alpha to help explore the impact of change in NFkB expression.*

Overall I was impressed by this analysis and do not have major technical concerns.

I do however have some presentational concerns. I think readers not used to this data would appreciate a single data point example of the an AE QTL - rather like Figure 2C, but also showing when the read out SNP is relative to the QTL SNP in potentially a cartoon.

As suggested, we added a cartoon in Figure 1 with two examples of allelic expression mapping in individual transcripts: in panel (A) is illustrated a proximal association for RAB31 and in panel (B) a more distal association for CACNA1E.

The more serious point is that it is clear that the authors struggled with the problem of both differential power and the inevitable point where the FDR cutoff means that a QTL might be present in many cell types but called in only 1 or a subset due to these issues. therefore the statement that most QTLs are shared is I think instinctively correct, but Figure 1A and Figure 1B visually read in the opposite direction. Given this I wonder if Figure 1A and 1B is really the way to present this information (though it does provide the numbers). I would encourage the authors to be much more up front about not just differential power in teh samples, but also the inherent problem of a continuum of effect sizes, and having to threshold each scenario. (Arguably one should use one of these new joint models that handles this statistically, such as from the Pritchard group, but I would not necessarily insist on this). Secondly I would question the Venn diagram presentation, perhaps focusing on the inverse - the number of QTLs with some clear cut single tissue specificity.

We thank the reviewer for the insightful comments and agree that approaches developed by Veyrieras, Pritchard and others are not straightforward to apply to allelic expression mapping data. To address the issues raised we have 1) revised presentation of results, including revision of Table 1 making now distinction between tissue-specific and population specific variation, in addition, to specifically address variation in sample sizes, 2) we also examined effects when power is similar across datasets.

- 1) *We replaced Figure 1A by the Table 1 to highlight the high proportion of mapped loci shared between tissues and populations and to clearly show that while tissues share much of variation and particularly in case of cultured cells, the population differentiation has much stronger diverging effect on cis-rSNPs than cell lineage. We removed Figure 1B from the new version of the manuscript.*
- 2) *To explore effect sizes and sharing in equally powered samples we ran bootstrapping analysis using 10 sets of 45 samples (80% of the smallest dataset (CEU LCLs)) randomly selected from each cell population. We focused on associations replicated in primary and now equally powered discovery datasets. The SNP – transcript pairs tested in the bootstrapping were based on original mapping in full datasets. We considered replication of primary cell population signal when at least one out of ten p-values was < 1% FDR and median of all p-values was lower than 10% FDR and then assessed if the replicated association showed tissue (population) sharing or tissue (population) specificity in original datasets. As now show in Figure E3, the primary associations replicated smaller datasets show similar patterns of tissue concordance, with primary purified monocytes having more distinct set of cis-regulatory SNPs than cultured fibroblasts or lymphoblasts. This removes question if the original differences were driven by ability to pick up weaker, tissue specific, signals in larger samples only. The equally powered datasets also allow us to show that YRI LCLs population with sequence divergence from the three other, Caucasian, sample panels shows consistently lower sharing of effects across cell panels, i.e. sequence context is quantitatively more important than cell lineage. Furthermore we show that even for population –and tissue shared sites variation the variance explained by such cis-rSNPs is similarly diverged in YRI panel suggesting that additional variants in YRI haplotypes may significantly alter the marginal effects showing consistency in among Caucasian panels. We note, and as indicated in our discussion, that we do map similar proportions of transcripts to SNPs as recent larger eQTL studies have achieved and thus, we believe that our observations are generalizable.*

Pertinent to this is I am not sure one can be quite so strong about the link between overall tissue expression levels and QTL - in some sense this is about the natural effect size vs the noise - in other words, a QTL as a proportion of the transcription, might be present at equal levels in low and high expressed genes, but only detectable compared to noise variation in the higher expressed case. The authors should soften this claim.

We agree the initial claim was too strong, and have removed this statement.

Other points:

I would prefer a consistent ordering of how the cell lines are listed in the paper text and the figures. There is a variety of orders. I would suggest doing this in ascending sample size (ie, Monocytes last).

As suggested, cell lines have now been consistently listed in order of ascending sample size: CEU LCLs, YRI LCLs, fibroblasts and monocytes.

I could not work out what the Y axis really was in Figure 6B. If it is like the other examples (eg, Figure 2C) why not use that? And label the axis.

The Y-axis represents the differential allelic expression between the two alleles of BLK and has now been labeled in Figure 5B.

There is a typo "GO accession number" should be "GEO accession number" I think.

This error has been corrected.

Reviewer #2: *Adoue et al. use allelic expression to characterize various features of genetic regulatory variation, from noncoding transcripts to GWAS relevance and transcription factor binding. As it is often the case in these kinds of descriptive studies where the aim is to characterize functional variation, there is no single hypothesis or one key conclusion. Their conclusions - of the*

pervasiveness, tissue sharing, and GWAS relevance of cis-regulatory variation - are rather incremental without tremendous novelty value, but especially the NFKP experiments to understand the role of TF binding in gene regulation are interesting. The study relies mostly on established methodology and focuses on biological discovery instead of novel computational approaches.

The paper touches many topics, and I had a hard time seeing how some of the analyses link to each other. It seems that the authors tried to think every single analysis that they could do with these data, and report all of them - instead of describing a well-defined set of analysis to answer a well-defined question. This is illustrated by the last paragraph of Introduction that is essentially a list of various topics that don't all have a lot to do with each other as biological phenomena, and this continues in Results that jumps from one thing to another. The authors should put substantial effort into building a proper narrative. Focusing on the TF binding part and maybe omitting some of the less novel analyses from the beginning of the paper (at least from the main text/figures) would make this paper more interesting to the readers.

Most of the conclusions that the authors present might be correct, but in the end I was unable to finish this review because the results and analysis are not described to a sufficient degree that I could really evaluate them. The sloppy reporting of the results is completely unacceptable. I got to page 9, wrote 2 pages of "minor comments", and stopped there. Reading the paper leaves me with a feeling that I hardly know what the analysis is, what the result is, and/or how significant it is. There are fold enrichments but no p-values, and vice versa. Several figures are lacking proper labeling. Results are frequently reported without any reference to the actual data and numbers. There are figures without references in the text. This is so frequent that I'm not even going to list all of them systematically (a pretty comprehensive list is below), and on page 9 I decided that this is beyond my job as a reviewer. I should not need to teach the authors - many of which have published quality work before - how to report data and results. They should have done this before even thinking of submitting this manuscript.

Please note that in the pull-down menu of different aspects of the manuscript, it was considerably difficult to say whether the quality of methods and conclusions was adequate or low - while my impression is that the methods and conclusions are probably OK, I cannot be sure of this because of the sloppy reporting, and thus I have no other option but to err on the cautious side.

We shortened the manuscript and removed all non-essential information and redundancies.

We also have improved the English language, as well as the overall flow of the manuscript to make it easier to read. All missing statistical tests used have been added with corresponding p-values.

Specific issues:

- When discussing lincRNAs, the authors need to consider a recent paper on the topic: PMID 24268656. Some of their results reported here as novel are already included in the Popadin et al. paper (as well as in Kumar et al. that the authors already refer to), and the results of that paper contradict some of the statements e.g. in the introduction (low rate of cis-regulation in lincRNAs is contrary to the Popadin et al. paper). In general, I'd suggest that the authors put substantially less emphasis in this section because few of the results are that novel, and it distracts the reader from the more important and interesting points.

As some of our results on lincRNAs were already reported in a very recent paper (Popadin et al.) as suggested by the reviewer, we significantly reduced our result section on non-coding RNA and improved our focus on the more novel aspects of the manuscript.

- Abstract: I believe that "bind" should be "bound" in "...motifs, predominantly bind by repressive factors..."

This error has been corrected.

- Introduction: "cis-regulatory mechanisms free of trans-acting variation and environmental effects" is a false statement. For example, the NFKP results in this paper are an elegant demonstration of how trans or environmental effects can modify cis-regulatory effects!

This sentence has been rephrased to: "Since both alleles are impacted by the same trans-acting and environmental effects, AE mapping reduces the complexity of gene expression to its cis components." (p.3).

- Introduction: "Studies of AE-mapping in lymphoblastoid cell-lines (LCLs) have revealed that approximately 30% of all loci have significant AE imbalance, with cis-rSNPs explaining more than

50% of the population variance in AE". Reference? How is this estimated in population samples - if it's a total across many samples, one would eventually reach 100% by sampling deeper (analogously to the number of variants discovered in sequencing studies), and if measured in an individual samples, this number is sensitive to technology and e.g. RNA-seq depth as well as statistical definition of AE.

We published data on AE mapping in LCLs (Ge et al, Nature Genetics, 2009), the reference has been added in the text (p.3). This is estimated across samples from the CEU LCL population. As hypothesized by the reviewer, the number of mapped allelically regulated transcripts by cis-variants will increase with the population size. For example, in this study we observed 70% of cis-regulated expressed transcripts in the monocyte population (n = 188) versus 30% for the CEU LCLs (n = 55).

- The number of samples used in this study is mentioned late in results. This is key information that should be in the abstract and early in the description of the study.

This information is now mentioned in the abstract (p.2).

- The authors need to specify in the methods and also briefly in the methods what exactly they are doing with introns and how they are assigning sites to transcripts (what are full and partial-length transcripts? What is "data from not conflicting regions"). This is a key point. If the authors are in any way annotating sites from the region of same gene into different transcripts and analyzing them independently, this is not appropriate. This is particularly relevant when reporting cis-rSNPs associating to several transcripts. The 35% of such SNPs seems like a high number to me, but since I don't know how transcripts are grouped, I can't really say if it makes sense or not.

As specified in the results and the methods sections, signal in introns has been used in the regression tests. This is particularly important, as 75% of the SNPs used for AE mapping were intronic. The description of full and partial-length transcripts has been added in the methods section (p.14): we decided to exclude intensity signal from genomic regions overlapped by more than one gene to avoid conflicting data. We termed "partial-length transcripts" transcripts for which we used only part of the AE data due to these low confidence regions. For "full-length" transcripts, no region was excluded. We used only one transcript per gene based on RNA-seq expression scores as specified in the methods section. We also systematically excluded overlapping transcripts from different genes. Therefore we are not analyzing the region of the same gene in different transcripts.

- In the methods, I assume that Ge et al. is the reference, but the method needs to be described briefly anyway. One should not need to look into a supplement of another paper to find out the basics of the analysis.

The method used for allelic-expression mapping is now described in more detail in the method section (p.13).

- Results "Through the simulation analysis we observed that the percentage of missing causal SNPs was below 5%..." What simulations? I don't find a description of these anywhere, not in this manuscript or in Ge et al.

We thank the reviewer for pointing out our error in omitting the description of simulations. We have now added description of simulations (Supplementary Methods) to test 1) accuracy of mapping strong, common allelic effects in varying haplotype structures (linkage disequilibrium) 2) impact of slight imputation error in 1000G, and 3) improvement of accuracy from combining data across populations for shared causal sites.

- All the barplots should have error bars

We can't add errors bar in all barplots since in graph bars represent a percentage of values within a population as in Figure 2C.

- Fig 1A: What are these numbers? "Associations" = number of transcripts, or something else?

Yes, these numbers represent the number of transcripts for which we mapped cis-rSNPs in each cell population. The legend of the Table 1 (representing now the number of mapped transcripts) has been changed to be more explicit.

- Fig 1B and reference to it in the text: where are the p-values? Also, it makes no sense to show H3K4me3 as bars and H3K4me1 as lines - they should both be bars, with different colors.

We removed this figure from the new version of the manuscript.

- Fig 1D: The weird 3D lines serve no other purpose than making it difficult to read the plot. Make it two-dimensional. Also, I think that the result shown in this figure has been seen so many times before that it could well be in the supplement.

This figure is now two-dimensional and in the supplementary section (Figure E6).

- Fig S4: Cutting the y-axis at 40-49% makes the difference look much more dramatic than what it actually is. Make it 0-49%.

The y-axis is now 0-49% as suggested.

- "This result [Fig 2A] suggests that protein-coding and non-coding transcripts shared same cis-regulatory variants." As far as I understand, these results don't say anything about the variants being the same, only that they have similar properties.

This is correct. We meant that the percentage of cis-regulated transcripts was similar between non-coding and protein-coding transcripts. This sentence has now been deleted in the text.

- The proportion of sharing between populations seems very low, which disagrees with earlier papers. How do the authors explain this?

To clarify sharing issues, we revised the presentation of results: Table 1 is now making distinction between tissue-specific and population specific variation. Sharing estimates included earlier both tissue and population variation, which are now separated. Sharing in cultured cells is high, whereas it is lower for cultured cells versus primary purified cells. Tissue and population divergence account for similar proportions of cis-variance. Globally the percentage of sharing is high between any two tissues (39-61%). In particular, Dimas et al. (Science, 2009) investigated eQTLs in fibroblasts and LCLs of 85 individuals and found 12.3% of sharing. Here, we found 35% of shared mapped transcripts between CEU LCLs (n=55) and fibroblasts (n=70), which is greater than that found by Dimas et al., and in agreement with our previously published study demonstrating the improved sensitivity of the AE mapping approach (Almlöf et al, PLoS ONE, 2012). Emilsson et al. (Nature, 2008) suggested over 50% of sharing (at an FDR of 1%) between two tissues using blood and adipose tissues as model. However this is based on the upper 25th percentile for differential expression. The percentage of sharing is close to 35% when using the four quartiles as in our study. Finally, our new mapping analysis that we carried out using equally sized datasets (see Supplementary Methods) gives us further validation for true differences in tissue and population variable cis-rSNPs as discussed above (see response to reviewer 1).

- "Among these cis-rSNPs representing 7% (n=24891) of all mapped cis-rSNPs, we observed a significant enrichment ($P=1 \times 10^{-9}$) of associations at intermediate distance (5kb to 200kb) from the TSS, when considering associations ranked 1 for each transcript only." Isn't this true almost by definition? I find it hard to imagine a scenario of multi-transcript associations where this isn't true - hence, this observation carries no biologically interesting information.

This sentence has been removed.

- "These observations argue for an impact of genetic variants on higher order chromatin function, as reported by us earlier for a single disease locus (Verlaan et al, 2009), which we now show to be a frequent phenomenon across human cell types." These results do not justify such a strong statement, and chromatin function is a very vague concept. The results here are suggestive at best without a more detailed analysis of the properties of the transcripts, and potentially also additional data of e.g. open chromatin in these regions.

This sentence has been rephrased to soften our conclusion: "We previously reported this type of effect for a single disease locus (Verlaan et al, 2009), and showed an impact of genetic variants on higher order chromatin function. Supplementary investigation would be needed to establish if this could be a common phenomenon across human cell types." (p.6).

- Is Fig S6 lacking legends or captions or something? What are these scores and bars of different colors?

These figures are original output from IPA (Ingenuity® Systems, www.ingenuity.com). A more explicit legend has been added (Figure E2).

- *Fig S7 is lacking axis labels, and there are more bars than labels.
This figure has been deleted.*
- *Fig 3A is a table, not a figure.
Figure 3A is now called Table II.*
- *Fig 3C is lacking an axis label.
Axis labels have been added to the figure.*
- *"Non-coding genes are rarely suggested as candidates for disease". This is not true - plenty of examples of ncRNAs with a putative role in disease are known, and this is widely recognized by the community.
This sentence has been deleted.*
- *Identification of cell-specific active motifs disrupted by cis-rSNPs : Where's the data? Fig 4a shows "selected" results - selected based on what? The full data needs to be provided.
Motifs with the greatest differences in cell specificity based on the heat map were selected for the Figure 3. The totality of the motif disruption data by cis-rSNPs is now accessible as supplementary data (Table E6).*
- *Fig 4A: what are these codes for the motifs? What do the numbers correspond to?
We exploited motifs defined by Neph et al. (Nature, 2012), which used DNaseI footprinting to discover 683 unique de novo motifs numbered 1 to 683. The name associated to each de novo motifs correspond to these numbers.*
- *"We also found that many de novo footprint-derived motifs are more specific to one of the three cell-types, suggesting selective activity for the associated transcription factors." Where's the data?
Some de novo footprint-derived motifs with cell-specificity are illustrated in Figure 3. Also, as mentioned above, the totality of the motif disruption data by cis-rSNPs is now accessible as supplementary data (Table E6).*
- *The section about de novo motifs needs to begin with a brief summary of what these motifs are. A reference to Neph et al. is not sufficient, because the nature of these motifs is critical for understanding the validity of the results.
The section on the de novo motif study called "Motif disruption at cis-rSNPs revealed new regulatory function" starts now with a brief description of the discovery of these de novo motifs (Neph et al. paper) (p.7).*
- *"We identified 110 motifs with significant allelic bias" out of how many? Fig 4b is also a table, not a figure. It needs to include some quantification of the bias in addition to the p-value.
We identified 110 motifs out of 1010 (TRANSFAC + de novo motifs) with significant allelic bias. The numbers of motifs used from the two databases have also been added in the text. Figure 4b is now referenced as Table III. Quantification of the bias has been added in Table III and Table E4.*
- *"we looked only at motifs with a predicted regulatory activity with a p-value below 0.01." What's this p-value?
For each motif, we performed a binomial test to assess for significant matrix score allelic bias: at cis-rSNPs location, motifs for transcriptional activators would be expected to more frequently exhibit a higher matrix score for the same haplotype as the over-expressed allele, while inhibitors should more frequently recognize the haplotype for the under-expressed allele. The p-value cut-off of 0.01 corresponds to the p-value of the binomial test. The text has been changed to clarify this point.*
- *I don't find any reference to Supplementary Figures 8 and 9.
The Supplementary Figure 8 is now mentioned in the text (Figure E8, p.7). The Supplementary Figure 9 has been deleted.*

Reviewer #3: *The authors present maps of allelic-expression (AE) associated SNPs using quantitative measurements of AE on Illumina Human1M BeadChips in different cell lines and populations, thereby extending work that has been published by Ge B et al. in Nature Genetics (2009) on the same platform. The authors further extend work on eQTLs published using different expression platforms (such as RNA-Seq, which has a different sensitivity towards detecting AE when compared to Illumina Human1M BeadChips). The authors use this platform to map common SNPs associated with cis regulation of both protein coding and non-coding genes ("cis-rSNPs"). There is a number of new observations described in this manuscript, facilitated by the sensitive parallel DNA/RNA genotyping approach used, that in my view will be of interest to MSB readers. The sensitivity of this platform enables the authors to comprehensively assess cis-rSNPs associated with non-coding RNA expression, which includes a number of SNPs that previously had been identified in GWAS studies. Amongst these there is an abundance of cis-rSNPs with inferred repressive effects. Additionally, by perturbing NFκB in lymphoblastoid cells the authors are able to identify a number of cis-regulated transcripts with altered AE upon perturbation, which shows the relevance of alerting/perturbing transcriptional networks for carrying out comprehensive AE analyses.*

My major criticism is rather a stylistic, than a scientific one. The manuscript is currently on the lengthy side, with lots of redundancy, and would significantly benefit both from English language editing and from streamlining. In some instances, colloquial expressions are being used by the authors were more scientific ones should be used for clarity (one example, on p. 5, the authors write that "protein-coding and non-coding transcripts shared same cis-regulatory variants", where I assume that what the authors meant to stress is the finding that the percentage of cis-regulated transcripts, as defined by their approach, was similar for those corresponding to non-coding vs. protein-coding transcripts.

As stated above, we shortened the manuscript and removed all non-essential information and redundancies. We also have improved the English language, as well as the overall flow of the manuscript to make it easier to read.

One additional more specific scientific comment is whether the authors can quantify to what extent their finding described on p. 6, i.e. that only 0.4% of rSNP associations were >200kb apart from the target transcript, may be affected by the low sensitivity of the BeadChip platform towards detecting associations involving distant sites (i.e., both RNA and DNA can contribute to the AE mapping with this parallel DNA/RNA genotyping approach, which may have significantly contributed to the peaking in identified cis-rSNPs for genic (vs non-genic) regions, displayed in Figure 1D).

Inaccurate phasing of DNA could in principle, hinder the mapping of distant marginal effects, but we do not think the BeadChip measurements of allelic biases are the cause of underestimating distant effects. We note that the theoretical impact of phasing inaccuracy appears small, for example in the fibroblast panel where we have family based samples and can accurately phase over long distances, the average distance of cell type (fibroblast) selective association from TES or TSS is 38kb and 1.1% of associations are greater than 200kb away. In case of monocytes where we rely on statistical phasing (no family information), the average distance of monocyte selective associations from TES or TSS is 23kb and 0.2% of associations are >200kb away from transcript boundaries. For 3 of 4 sample panels we used family based phasing and believe that there's no underestimation of distal association densities. To acknowledge this possibility in the case of monocytes, we have now added the sentences (p.6, 1st paragraph): "The density of long-range associations may be slightly underestimated for the monocyte sample due to reliance on statistical rather than family-based approach in phase assignment and potential of confounding errors in long-range haplotypes. We observe rate of long-range effects of 0.2% in monocytes versus 1.1% in the three other cell types. "

Thank you again for submitting your work to Molecular Systems Biology. We have now heard back from reviewer #2 who accepted to evaluate the revised study. As you will see, this referee acknowledge that this revised manuscript has been improved and recognize that the topic of the study is interesting. However, this reviewer still raises significant issues with regard to data presentation and the way the results are reported.

As you may know, our editorial policy is to allow a single round of revision only. The issues raised by this reviewer are important. It seems however that they are potentially addressable. We would thus ask you to go very carefully through all the points listed by the reviewer and address them convincingly in an exceptional and last round of revision. The issues still relate to the need of a more rigorous presentation of the results (eg label properly graphs, include effect size in addition to p-value, etc...). A potentially more serious issue is the multiple-testing problem that appears to affect the motif analysis. The very low level of association sharing across all tissue should also be investigated as this appears highly abnormal to reviewer #2.

Reviewer #2:

Adoue et al. have revised their paper with some degree of success - the manuscript is much more focused and now tells a coherent, interesting story. However, in my first review I pointed out that missing statistical description all over the paper was a major issue, and the authors have not addressed these concerns by systematically revising the paper. The same problems are still abundant, without any explanation of why my comments were ignored. It is really an interesting study, and I really wish that the authors would do it justice by doing careful work in reporting it. I'm listing specific issues below.

- I am astonished to still find numerous figures with missing axis labels. Fig. E3a, E4, E5, E6, etc - even though I pointed this out before, and it should be clear to everyone without saying.
- Previously, I asked the authors to add error bars to plots, and they answered "We can't add error bars in all barplots since in graph bars represent a percentage of values within a population as in Figure 2C." Of course confidence intervals can be calculated for proportions/percentages (see e.g. `binom.test` in R). Google would have told them this. Please add the error bars.
- I asked the authors need to describe in each and every case what they are testing, what are the resulting test statistic values (e.g. 2.5x enrichment), and what are the p-values and statistical tests. The authors have added p-values and statistical tests, but ignored everything else that I asked them to do. This is not acceptable. In genomics, data is often so large that even tiny effects are statistically extremely significant, but in order to estimate the real magnitude of the biological signal, the reader needs to know if e.g. an enrichment is 8x or 1.05x. If there is no corresponding figure/table giving these data, they need to be included in the text. For example: "x2 enrichment of GWAS hits in group A compared to group B, t-test p-value=0.01".
- Variant sharing: In the sharing percentages, do the authors only count genes that have been analyzed in the other population? They should do this - otherwise the proportions don't measure tissue-specificity of regulatory variants but specificity of gene expression.
- The ubiquitous sharing proportion of 3.7% is contrary to all literature - it should be closer to 50% - and there must be something wrong. This should be discussed and analyzed explicitly. Monocytes might be outliers due to being primary tissues, but not this much.
- "...disrupted binding sites per motif at cis-rSNPs based on cell-type specificity (Table E4)." I can't find a file titled E4 (might be the journal website's fault), and Data Set 3 file is the only one that seems to make sense. However, I don't see any stats of cell type specific effects there.
- "We identified 110 motifs (11%) with significant matrix score allelic bias (P binomial test < 0.05) (Table III and Table E4)." In Data Set 3, assuming that this is the right table (the p-values correspond to those in Table III so I guess it is), I'm getting 144 (and not 110) motifs with $p < 0.05$ - where's the discrepancy?
- In the motif analysis, I apologize for not picking up an important point before: nominal $p < 0.05$ is

not the proper way to claim significance when 1380 tests are done, and the authors need to account for multiple testing. There's hardly anything that would be significant after a Bonferroni correction ($0.05/1380 = 3.623188e-05$), but that's too stringent and an FDR type of approach seems to work better. The authors should choose a p-value threshold that corresponds to a chosen FDR, or at the very least report the FDR of their chosen threshold (for $p < 0.05$, it seems to be around 50% which is way too high). To get to 10% FDR (with $p < 0.001$ or thereabouts) the authors are left with only a handful of significant motifs. Many of examples listed in the text are far from significant, and substantial revisions are needed to account for this.

- The authors have not addressed my previous comment "Identification of cell-specific active motifs disrupted by cis-rSNPs : Where's the data? Fig 4a shows "selected" results - selected based on what?" The authors present some motifs in Fig. 3 and Table III, but I don't find a description in the paper of HOW the selection was done, and their response to me "the greatest differences in cell specificity based on the heat map" is not a proper statistically defined set of criteria. Also, Fig. 3 should have a proper scale - what is high and what is low?

Minor comments

- Describing Fig. 1, the authors use proximal/distal terminology for cis variants that are closer or farther away from TSS. However, proximal/distal is often used for cis/trans eQTLs - I suggest revising the text to avoid misunderstandings.

- The sharing statistics in Table 1 are very confusing. Instead of percentages, the authors should give both percentages and absolute numbers, and indicate clearly where the numbers come from and what's the denominator. I think I can figure out that "total number of shared associations" represents from the main text, I assume that the authors mean "Associations shared with ≥ 1 other sample"? I assume that "% of shared associations across tissues and populations -- 6.18% YM" means that X/8175 (6.18%) of monocyte associations are found in YRI (it took me a few minutes to figure this out, plus a calculator to get some idea of the absolute numbers that this corresponds to). Why don't the authors calculate the reciprocal number for YRI associations found in monocytes (and fibroblasts)?

- I asked about the numbering and codes for motifs "Fig 4A: what are these codes for the motifs? What do the numbers correspond to?" and the authors explain in the response to me where the data comes from. Do they not think that this explanation should be included in the figure legend so that readers can also understand what's being plotted?

- Page 9: what's "Supplementary IV"?

- "The intersection of cis-rSNPs with variants from the GWAS catalog reveals a large number of potential causal variants" Since the authors don't have genome sequencing data, they cannot know if their top variants are actually causal. What they are revealing is potential causal regulatory mechanisms underlying the associations - the precise causal variant is another question. Please revise the text.

- Discussion: "We noticed an enrichment of lincRNAs or antisense transcripts in genes regulated by NF B." I don't see this result discussed anywhere else in the paper. It should be either included in Results (with proper statistical description) or dropped from Discussion.

2nd Revision - authors' response

18 August 2014

Point-by-point response to the referee:

Reviewer #2: *Adoue et al. have revised their paper with some degree of success - the manuscript is much more focused and now tells a coherent, interesting story. However, in my first review I pointed out that missing statistical description all over the paper was a major issue, and the authors have not addressed these concerns by systematically revising the paper. The same problems are still abundant, without any explanation of why my comments were ignored. It is really an interesting study, and I really wish that the authors would do it justice by doing careful work in reporting it. I'm listing specific issues below.*

- I am astonished to still find numerous figures with missing axis labels. Fig. E3a, E4, E5, E6, etc - even though I pointed this out before, and it should be clear to everyone without saying.

We apologize for the previously missing axis labels. These have now been carefully added in Figures E3-E6 and Figure E8.

- Previously, I asked the authors to add error bars to plots, and they answered "We can't add errors bar in all barplots since in graph bars represent a percentage of values within a population as in Figure 2C." Of course confidence intervals can be calculated for proportions/percentages (see e.g. binom.test in R). Google would have told them this. Please add the error bars.

As requested by the reviewer, we have now added error bars in all graph bars: Figure 2C, 4, E1, E3, and E8.

- I asked the authors need to describe in each and every case what they are testing, what are the resulting test statistic values (e.g. 2.5x enrichment), and what are the p-values and statistical tests. The authors have added p-values and statistical tests, but ignored everything else that I asked them to do. This is not acceptable. In genomics, data is often so large that even tiny effects are statistically extremely significant, but in order to estimate the real magnitude of the biological signal, the reader needs to know if e.g. an enrichment is 8x or 1.05x. If there is no corresponding figure/table giving these data, they need to be included in the text. For example: "x2 enrichment of GWAS hits in group A compared to group B, t-test p-value=0.01".

We have now included effect size in addition to p-value for all performed tests in which there is no figure or table already providing this information: see last paragraph from pages 5-6, and pages 8-10.

- Variant sharing: In the sharing percentages, do the authors only count genes that have been analyzed in the other population? They should do this - otherwise the proportions don't measure tissue-specificity of regulatory variants but specificity of gene expression.

We assessed sharing by analyzing all transcripts mapped in at least one tissue (n=10900) in all others. The allelic expression method sensitivity allows for mapping of transcripts considered as not expressed (FPKM<0.01). This is particularly interesting for the non-coding RNA often weakly expressed. As suggested by the reviewer, sharing percentage provided in the manuscript could be bias by the specificity of gene expression. We then tested a restricted list of transcripts expressed in all cell types (n=1886) and mapped in at least one tissue. The proportion of sharing was very similar between the two lists: 22.8% and 21.4% of CEU LCLs SNP-transcripts associations, 19.2% and 17.9% of Fibroblasts, and 9.5% and 9.5% of Monocytes from full and expression-restricted datasets respectively were shared with the two others tissues. All others sharing and cell-specific percentages between the two lists of associations were as similar. As the numbers were very close, and the full dataset contains ~6 fold more transcripts we decided to keep sharing information from all transcripts.

- The ubiquitous sharing proportion of 3.7% is contrary to all literature - it should be closer to 50% - and there must be something wrong. This should be discussed and analyzed explicitly. Monocytes might be outliers due to being primary tissues, but not this much.

To address this concern we have clarified our approach and performed new analyses presented in the paper and detailed below.

We have now emphasized that our approach to calculate sharing across tissues was designed to address exactly mapped marginal associations between tissues given that our emphasis is on finding causal cis-variants. To this effect we have added a sentence preceding the introduction of Table 1 (reporting tissue sharing) on page 5 (paragraph 2): "We applied a stringent approach to assess exactly shared top associations across tissues requiring not only significant association in both tissues, but also converging association pattern." In line with this the ubiquitous sharing of 3.7% represents the proportion of transcripts mapped in the 4 cell populations and across the total number of mapped transcripts. This low sharing is then in part explained by the fact that the number of SNP-gene associations is high in monocytes. We now report the sharing between the different tissues starting from each tissue (Table E3). We identified 10 to 23% of SNP-transcript associations in each tissue (same population) as shared in the two others.

To allow comparison of our approach to recent works reporting higher level of sharing we are now presenting a parallel analysis as shown recently for eQTL sharing based on excess of low p-values in second tissue among the sites mapped in first (PMID: 22941192).

Specifically, we took lead association in one tissue (based on our definition by FDR) and fetch exactly the data for same SNP-transcript pair in the other tissues. We used as input these p-value lists in the R package "qvalue" and run the default setting in the two data sets separately. The output π_0 in the summary file is used to calculate π_1 values ($1-\pi_0$), which represents the proportion of shared hits among the tested. This approach is more liberal and allows for uncertainties of mapping accuracy in differently powered datasets, consequently pairwise sharing estimated based on this approach is substantially higher (40 – 60%). We have added the following description in results section (page 5, paragraph 2) and included the table below in the supplement: "We noted that our tissue sharing is conservative as compared to methods recently used to estimate tissue sharing in eQTL studies (PMID: 22941192) allowing for uncertainty in mapping accuracy. Estimates of pairwise tissue sharing based on π_1 values are similar to eQTL studies and range from 39% between YRI LCLs and Fibroblasts to over 60% for monocyte lead associations and the three other sample panels (Table E4)(see Methods)."

Table E4. Pairwise sharing estimated based on π_1 values

	Leading population			
	CEU	YRI	FB	MNC
CEU	0.9998	0.4060	0.4445	0.6114
YRI	0.5111	0.9995	0.4968	0.6411
FB	0.4534	0.3906	0.9995	0.6317
MNC	0.5684	0.4705	0.6161	0.9995

- "...disrupted binding sites per motif at cis-rSNPs based on cell-type specificity (Table E4)." I can't find a file titled E4 (might be the journal website's fault), and Data Set 3 file is the only one that seems to make sense. However, I don't see any stats of cell type specific effects there.

The summary of the number of disrupted binding sites per motif based on cell-type specificity is available in Table E7. Big excel files need to be downloaded as datasets and explain the discrepancy. For each motif, significant differences in the number of disrupted transcription factor binding sites per tissue are calculated based on Chi-squared test with p-value < 0.05.

- "We identified 110 motifs (11%) with significant matrix score allelic bias (P binomial test < 0.05) (Table III and Table E4)." In Data Set 3, assuming that this is the right table (the p-values correspond to those in Table III so I guess it is), I'm getting 144 (and not 110) motifs with $p < 0.05$ - where's the discrepancy?

We identified 144 motifs with significant matrix score allelic bias (P binomial test < 0.05). The number of 110 was a typo. Using a Bonferroni correction approach (see next comment), we are now reporting the identification of 130 and 63 motifs using p-value binomial test < 0.05 and < 0.01 respectively.

- In the motif analysis, I apologize for not picking up an important point before: nominal $p < 0.05$ is not the proper way to claim significance when 1380 tests are done, and the authors need to account for multiple testing. There's hardly anything that would be significant after a Bonferroni correction ($0.05/1380 = 3.623188e-05$), but that's too stringent and an FDR type of approach seems to work better. The authors should choose a p-value threshold that corresponds to a chosen FDR, or at the very least report the FDR of their chosen threshold (for $p < 0.05$, it seems to be around 50% which is way too high). To get to 10% FDR (with $p < 0.001$ or thereabouts) the authors are left with only a handful of significant motifs. Many of examples listed in the text are far from significant, and substantial revisions are needed to account for this.

In the motif analysis, we used default p-value ($P < 1 \times 10^{-4}$) from the FIMO software to detect sites for transcription factor binding and not $P < 0.05$ as reported by the reviewer. However as we did not account for multiple testing, we added a Bonferroni correction ($0.05/1380 = 3.62e-05$). Using this approach, we restrict the motif analysis to the 45% most significant sites that we reported in the previous version of the manuscript. The conclusions remain unchanged even with more stringent statistical handling of data. The main example from Figure 4C-E remains significant. The example from the supplementary data was deleted. All the analysis based on the motif discovery were updated: Figure 3,

Figure 4, Table III, Table E6, Table E7 and Table E8. We also edited the text related to this section (from page 7 - paragraph3, to page 8 - paragraph 2).

- The authors have not addressed my previous comment "Identification of cell-specific active motifs disrupted by cis-rSNPs : Where's the data? Fig 4a shows "selected" results - selected based on what?" The authors present some motifs in Fig. 3 and Table III, but I don't find a description in the paper of HOW the selection was done, and their response to me "the greatest differences in cell specificity based on the heat map" is not a proper statistically defined set of criteria. Also, Fig. 3 should have a proper scale - what is high and what is low?

We performed a new identification of cell-specific active motifs using the motifs remaining after the Bonferroni correction. We summarized the number of each motif that were disrupted by a cis-rSNP and in each cell-type. For the selection of cell-type specific motifs, we performed a Chi-squared test between cell-types (see previous comment). Any motif with a p-value below 0.05 was considered as cell-specific. These data are now available in Table E7.

The scale of Figure 3 is now going from -1.2 to 1.2. This scale is based on matrix2png software, with normalization to a mean value of 0 and variance 1 for each row.

Minor comments

- Describing Fig. 1, the authors use proximal/distal terminology for cis variants that are closer or farther away from TSS. However, proximal/distal is often used for cis/trans eQTLs - I suggest revising the text to avoid misunderstandings.

We edited the text as suggested. The new legend of Figure 1 is:

“(A) Example of cis-rSNP located close to the TSS of its associated transcript. Differential AE of RAB31 is associated to rs1893126 genotype. (B) Example of cis-rSNP located far away from its associated transcript.”

- The sharing statistics in Table 1 are very confusing. Instead of percentages, the authors should give both percentages and absolute numbers, and indicate clearly where the numbers come from and what's the denominator. I think I can figure out that "total number of shared associations" represents from the main text, I assume that the authors mean "Associations shared with ≥ 1 other sample"? I assume that "% of shared associations across tissues and populations -- 6.18% YM" means that X/8175 (6.18%) of monocyte associations are found in YRI (it took me a few minutes to figure this out, plus a calculator to get some idea of the absolute numbers that this corresponds to). Why don't the authors calculate the reciprocal number for YRI associations found in monocytes (and fibroblasts)?

As suggested, we are now reporting both percentages and absolute numbers for the sharing statistics in Table 1. We indicate more clearly the different numbers by a legend below Table 1 and report all combination of sharing in Table E3.

- I asked about the numbering and codes for motifs "Fig 4A: what are these codes for the motifs? What do the numbers correspond to?" and the authors explain in the response to me where the data comes from. Do they not think that this explanation should be included in the figure legend so that readers can also understand what's being plotted?

As suggested we have included in the legend of Figure 3 an explanation of the naming of the de novo motifs: "Numbers correspond to the unique de novo motifs numbered 1 to 683 as defined by Neph et al. (see main text)".

- Page 9: what's "Supplementary IV"?

We have fixed this typo and replaced "Supplementary IV" by "Table E9". This table contains a summary of the transcripts perturbed by NFkB.

- "The intersection of cis-rSNPs with variants from the GWAS catalog reveals a large number of potential causal variants" Since the authors don't have genome sequencing data, they cannot know if their top variants are actually causal. What they are revealing is potential causal regulatory mechanisms underlying the associations - the precise causal variant is another question. Please revise the text.

This sentence has been rephrased to: "The intersection of cis-rSNPs with variants from the GWAS catalog reveals a large number of shared hits".

- Discussion: "We noticed an enrichment of lincRNAs or antisense transcripts in genes regulated by NFκB." I don't see this result discussed anywhere else in the paper. It should be either included in Results (with proper statistical description) or dropped from Discussion.

This result was reported at the end of the result section page 10. The sentences in the result and discussion sections have been rephrased:

- "Interestingly, we observed among the mapped and perturbed genes a significant enrichment of lincRNAs and processed transcripts (~2 fold, Chi-squared test, $P = 4.4 \times 10^{-20}$)."

- "We noticed an enrichment of lincRNAs or processed transcripts in genes regulated by NFκB."

3rd Editorial Decision

31 August 2014

Thank you again for submitting your work to Molecular Systems Biology. We are now globally satisfied with the modifications made and will be able to accept your manuscript for publication pending the following minor modifications:

- Please include the accession numbers for all the datasets in the data availability section. It would be ideal if you would separate the data availability section in two sub-sections entitled "Referenced Data" to list the pre-existing data and "Primary Data" to list the datasets that are unique to this study:

Primary datasets:

- the allelic-expression data on the Yoruban cohort of lymphoblastoid cells (LCLs)
- the RNA-seq. data on primary monocytes
- the RNA-seq. data in Yoruban LCLs
- the H3K4me1 and H3K4me3 ChIP-seq data on primary monocyte samples (x2)
- all the NFκB perturbation data followed by genome-wide allelic-expression data generation with Caucasian and Yoruban LCLs with control and treated samples in multiple replicates

Referenced datasets:

- The allelic expression data on fibroblasts was used in methylation - expression correlation in paper by Wagner et al. (2014)
- The allelic expression on monocytes] without our present steps of filtering, transcript selection based on RNA-seq. data overlaps with Alml^f et al. (doi: 10.1371/journal.pone.0052260)
- The ChIP-seq data on LCLs and fibroblasts are used Light et al., where they were used to verify a new method for allelic effects on chromatin (Light et al. Epigenetics, 2014)

- Please cite and reference Wagner et al 2014 and Light et al 2014.

- there are still instances where the fold enrichment are not indicated: eg page 8 "...significant enrichment (t-test, $P = 5.4 \times 10^{-7}$) for both chromatin marks on the same haplotype..." and page 9 "...we observed significantly (Chi-squared test, $P = 4 \times 10^{-17}$) higher NFκB signals at cis-rSNP locations...".

3rd Revision - authors' response

09 September 2014

Response to comments:

- Please include the accession numbers for all the datasets in the data availability section. It would be ideal if you would separate the data availability section in two sub-sections entitled "Referenced Data" to list the pre-existing data and "Primary Data" to list the datasets that are unique to this study:

As requested, we edited the data availability section as follow:

“Primary Data:

cDNA and gDNA raw data for YRI LCLs can be accessed through the GEO accession number GSE52442.

RNA-seq data for primary monocytes and YRI LCLs can be access through the GEO accession number GSE53837.

ChIP-seq data for primary monocytes can be access through the GEO accession number GSE53837.

cDNA and gDNA raw data after NFkB perturbation (treatment and controls) for the two LCLs trio can be accessed through the GEO accession number GSE61254.

Referenced Data:

cDNA and gDNA raw data for monocytes and fibroblasts can be accessed through accession number EGAS0000000119 at EGA and the GEO accession number GSE52442, respectively.

RNA-seq data for fibroblasts can be access through the GEO accession number GSE53837.

ChIP-seq for fibroblasts and LCLs can be access through the GEO accession number GSE53837.”

- Please cite and reference Wagner et al 2014 and Light et al 2014.

We added the references for Wagner et al 2014 and Light et al 2014 in the text page 4 paragraph 3 and page 8 paragraph 3, respectively.

- there are still instances where the fold enrichment are not indicated: eg page 8 "...significant enrichment (t-test, $P = 5.4 \times 10^{-7}$) for both chromatin marks on the same haplotype..." and page 9 "...we observed significantly (Chi-squared test, $P = 4 \times 10^{-17}$) higher NFkB signals at cis-rSNP locations...".

We have added 3 fold enrichments and the text is as follow:

Page 8, paragraph 2:

"...significant enrichment (~ 1.2 fold, t-test, $P = 5.4 \times 10^{-7}$) for both chromatin marks on the same haplotype..." .

"... leading to the disruption of a “repressor” site, we observed the opposite trend (~ 1.2 fold, t-test, $P = 3.2 \times 10^{-4}$, Figure 4B)..." .

Page 9, paragraph 2:

"...we observed significantly (~ 1.1 fold, Chi-squared test, $P = 4 \times 10^{-17}$) higher NFkB signals at cis-rSNP locations..." .